# Dysfunction of Calcyphosine-Like gene impairs retinal angiogenesis through the MYC axis and is associated with familial exudative vitreoretinopathy

Wenjing Liu[1,2,3†], Shujin Li[1,2†], Mu Yang[1,2†], Jie Ma[1†], Lu Liu[1], Ping Fei[4], Qianchun Xiang[1], Lulin Huang[1], Peiquan Zhao[4]*, Zhenglin Yang[1,2,5,6]*, Xianjun Zhu[1,2,3,5,6]*

[1]The Sichuan Provincial Key Laboratory for Human Disease Gene Study, Center for The Sichuan Provincial Key Laboratory for Human Disease Gene Study, Center for Medical Genetics, Sichuan Provincial People's Hospital, University of Electronic Science and Technology of China, Chengdu, China; [2]Center for Natural Products Research, Chengdu Institute of Biology, Chinese Academy of Sciences, Chengdu, China; [3]Henan Branch of National Clinical Research Center for Ocular Diseases, Henan Eye Hospital, People's Hospital of Zhengzhou University, Henan Provincial People's Hospital, Zhengzhou, China; [4]Department of Ophthalmology, Xinhua Hospital Affiliated to Shanghai Jiao Tong University School of Medicine, Shanghai, China; [5]Jinfeng Laboratory, Chongqing, China; [6]Research Unit for Blindness Prevention of Chinese Academy of Medical Sciences (2019RU026), Sichuan Academy of Medical Sciences and Sichuan Provincial People's Hospital, Chengdu, China

*For correspondence:
zhaopeiquan@xinhuamed.com.cn (PZ);
yangzhenglin@cashq.ac.cn (ZY);
xjzhu2@126.com (XZ)

†These authors contributed equally to this work

**Abstract** Familial exudative vitreoretinopathy (FEVR) is a severe genetic disorder characterized by incomplete vascularization of the peripheral retina and associated symptoms that can lead to vision loss. However, the underlying genetic causes of approximately 50% of FEVR cases remain unknown. Here, we report two heterozygous variants in calcyphosine-like gene (*CAPSL*) that is associated with FEVR. Both variants exhibited compromised CAPSL protein expression. Vascular endothelial cell (EC)-specific inactivation of *Capsl* resulted in delayed radial/vertical vascular progression, compromised endothelial proliferation/migration, recapitulating the human FEVR phenotypes. *CAPSL*-depleted human retinal microvascular endothelial cells (HRECs) exhibited impaired tube formation, decreased cell proliferation, disrupted cell polarity establishment, and filopodia/lamellipodia formation, as well as disrupted collective cell migration. Transcriptomic and proteomic profiling revealed that *CAPSL* abolition inhibited the MYC signaling axis, in which the expression of core MYC targeted genes were profoundly decreased. Furthermore, a combined analysis of *CAPSL*-depleted HRECs and *c-MYC*-depleted human umbilical vein endothelial cells uncovered similar transcription patterns. Collectively, this study reports a novel FEVR-associated candidate gene, *CAPSL*, which provides valuable information for genetic counseling of FEVR. This study also reveals that compromised CAPSL function may cause FEVR through MYC axis, shedding light on the potential involvement of MYC signaling in the pathogenesis of FEVR.

## eLife assessment

This study explores the role of calcyphosine-like (CAPSL) in Familial Exudative Vitreoretinopathy (FEVR) via the MYC pathway, offering **valuable** insights into disease mechanisms that are supported

by a solid, multi-pronged approach. The manuscript, which presents the phenotype of an interesting new mouse model, provides **convincing** evidence that CAPSL variants cause disease.

## Introduction

A well-organized vascular system is crucial for the proper development of most tissue and organ morphogenesis (*Vogenstahl et al., 2022*; *Ye et al., 2009*). Consequently, angiogenic defects have been associated with several congenital human diseases (*Carmeliet and Jain, 2011*; *Fallah et al., 2019*). Familial exudative vitreoretinopathy (FEVR) is a heritable vitreoretinopathy characterized by defective retinal angiogenesis and various complications, such as extensive neovascularization, exudation, retinal folds and detachments, and vision loss (*Yonekawa et al., 2015*). Based on an ocular screening of over 60,000 Chinese newborns, the estimated prevalence of FEVR is around 0.46% (*Fei et al., 2021*). Due to the significant genetic heterogeneity, FEVR displays all Mendelian forms of inheritance: autosomal dominant (AD), autosomal recessive (AR), or X-linked recessive (XR) (*Plager et al., 1992*; *Gow and Oliver, 1971*; *de Crecchio et al., 1998*). To date, mutations in 17 genes and 1 locus have been identified to cause FEVR, including Norrin (*NDP*) (*Chen et al., 1993*), frizzled 4 (*FZD4*) (*Robitaille et al., 2002*), low-density lipoprotein receptor-related protein 5 (*LRP5*) (*Gong et al., 2001*; *Jiao et al., 2004*), low-density lipoprotein receptor-related protein 6 (*LRP6*) (*Li et al., 2022*), tetraspanin-12 (*TSPAN12*) (*Junge et al., 2009*; *Nikopoulos et al., 2010*), α-catenin (*CTNNA1*) (*Zhu et al., 2021*), β-catenin (*CTNNB1*) (*Panagiotou et al., 2017*; *Liu et al., 2024*; *He et al., 2023*), p120-catenin (*CTNND1*) (*Yang et al., 2022*), zinc finger protein 408 (*ZNF408*) (*Collin et al., 2013*), kinesin family member 11 (*KIF11*) (*Robitaille et al., 2014*), atonal homolog 7 (*ATOH7*) (*Khan et al., 2012*), exudative vitreoretinopathy 3 (*EVR3*) (*Downey et al., 2001*), integrin-linked kinase (*ILK*) (*Park et al., 2019*), jagged canonical Notch ligand 1 (*JAG1*) (*Zhang et al., 2020*), discs large MAGUK scaffold protein 1 (*DLG1*) (*Zhang et al., 2021*), transforming growth factor beta receptor 2 (*TGFBR2*) (*Asano et al., 2021*), sorting nexin 31 (*SNX31*) (*Xu et al., 2023*), and ER membrane protein complex subunit 1 (*EMC1*) (*Li et al., 2023*). Nevertheless, these mutations can explain only approximately 50% of FEVR cases (*Salvo et al., 2015*; *Kashani et al., 2014*).

Calcyphosine-Like (*CAPSL*) is a protein-coding gene whose function remains elusive. In 2019, Julia Schreml pointed out that *CAPSL* is a promising candidate gene for multiple symmetric lipomatosis (MSL), a rare adipose tissue disorder of largely unknown etiology (*Lindner et al., 2019*). Nevertheless, the specific pathogenic mechanism by which CAPSL triggers MSL remains unclear. The molecular mechanisms governing cellular processes and signaling cascades regulated by CAPSL have yet to be elucidated.

*MYC* gene family is a group of genes most widely investigated and implicated in the formation, maintenance, and progression of several different cancer types, such as breast cancer, lymphoma, and prostatic cancer (*Dang, 2016*; *Beaulieu et al., 2020*; *Carroll et al., 2018*; *Sander et al., 2012*). MYC is a master regulator in MYC signaling, which participates in cell metabolism, proliferation, differentiation, and migration (*de Alboran et al., 2001*). The important role of *Myc* in vascularization and angiogenesis during tumor development has already been reported in mouse models (*Baudino et al., 2002*; *Knies-Bamforth et al., 2004*). Vascular endothelial cell (EC)-specific knockout of *Myc* in mice led to impaired vascular expansion, thinned and poorly branched vascular, and reduced endothelial proliferation (*Wilhelm et al., 2016*). Conversely, EC-specific *Myc* overexpression caused sustained vascular outgrowth, increased EC proliferation, and vessel density (*Wilhelm et al., 2016*). These studies indicate that MYC is a regulating factor in coordinating EC behavior and vessel development.

In the current study, from a large cohort of FEVR patients, we identified a missense mutation and a stop-gain mutation in the *CAPSL* gene from four FEVR patients by whole-exome sequencing (WES) analysis. Furthermore, EC-specific *Capsl*-knockout mouse model exhibited FEVR-like retinal vascular defects, emphasizing the indispensable role of *Capsl* for the retinal vascular architecture. Knockdown of *CAPSL* led to compromised proliferation of stalk ECs and retarded migration of tip ECs by regulating the MYC signaling axis. Collectively, this study not only identifies *CAPSL* as a candidate gene for FEVR, but also presents that a compromised MYC signaling axis might contribute to FEVR.

## Results

### WES analysis revealed two *CAPSL* variants in FEVR patients

To evaluate the causative variants that account for FEVR, we applied WES on genomic DNA samples from 120 FEVR families without mutations in known FEVR genes. Sanger sequencing was further applied to validate the variants and genotype–phenotype co-segregation analysis. Thus, two novel candidate variants in the *CAPSL* gene (NM_144647) were identified in patients with FEVR, and the variants were predicted to be pathogenic by Mutation taster, Polyphen-2, and PROVEAN (*Supplementary file 1*). In family 3036, a heterozygous c.88C>T (p.R30X) variant was identified in an infant (age 0–5 years old) (*Figure 1A*). His father was a heterozygous carrier and manifested vascular defects in the retina, including peripheral neovascularization area and leakage by fundus fluorescein angiography, whereas the wild-type mother showed normal vision (*Figure 1B, D*). The affected amino acid is highly conserved among different species (*Figure 1C*). The proband (II:1) in the 3104 family has been diagnosed with FEVR and identified with a heterozygous variant c.247C>T (p.L83F) (*Figure 1A*). His father has also been identified as a carrier of this heterozygous variant, manifested with FEVR symptoms, according to the medical records. Nevertheless, clinical examination data are presently unavailable.

To determine the pathogenesis of *CAPSL* variants for FEVR, we first investigated the effect of variants on the expression of *CAPSL*. Wild-type and mutant coding sequences of *CAPSL* were introduced to the GFP-tagged pcDNA3.1 vector, and an IRES2 box was inserted to prevent the fusion expression of CAPSL and GFP. Immunoblot analysis of cells transfected with plasmids revealed that the CAPSL-R30X protein was undetectable, and the L83F variant resulted in a considerably reduced protein level (*Figure 1E, F*). Conversely, neither variant influenced the mRNA levels of *CAPSL* (*Figure 1G*). CAPSL is localized in both the cytoplasm and the nucleus of cells (https://www.proteinatlas.org/), and we then further verified whether these variants have any impact on CAPSL localization. Immunocytochemistry analysis revealed that, consistent with immunoblot analysis, CAPSL-R30X protein was undetectable, and CAPSL-L83F mutant protein exhibited no significant change in localization (*Figure 1—figure supplement 1*). The TGA stop codon can also have an effect on alternative splicing in certain scenarios, and bioinformatics predictions suggested that the impact of *CAPSL* 88C>T variant on alternative splicing is minimal (*Figure 1—figure supplement 2*). Thus, these variants may cause FEVR by reducing CAPSL protein abundance.

### Depletion of *Capsl* in vascular ECs caused FEVR-liked phenotypes

We generated an inducible EC-specific *Capsl*-knockout mouse model by breeding mice carrying a loxp-flanked allele of *Capsl* with tamoxifen-inducible *Pdgfb-iCreER* (*Claxton et al., 2008*) transgenic animals to determine whether depletion of *Capsl* in mouse ECs could cause FEVR-like phenotypes (*Figure 2—figure supplement 1A*). Sanger sequencing analysis confirmed that all experimental mice did not harbor confounding mutations such as *rd1* and *rd8* mutations (*Figure 2—figure supplement 2*). *Capsl*^loxp/loxp^, *Pdgfb-CreER* mice (hereafter named iECKO), *Capsl*^loxp/+^, *Pdgfb-CreER* mice (hereafter named iECKO/+), and their control littermates (*Capsl*^loxp/loxp^ and *Pdgfb-CreER*, hereafter named Ctrl) were induced by consecutive intraperitoneal injection of 50 µg tamoxifen per pup at postnatal day 1 (P1) to P3 before sacrifice (*Figure 2—figure supplement 1B, C*). iECKO mice showed indistinguishable overall appearance compared to their Ctrl littermates. The efficiency of EC-specific deletion of *Capsl* was confirmed by real-time quantitative PCR (RT-qPCR) and western blot analysis of P35 mouse lung tissues. The results showed a pronounced reduction in CAPSL expression, at both mRNA and protein levels, in iECKO mice compared to Ctrl littermates (*Figure 2—figure supplement 1D*).

FEVR is a genetically heterogeneous eye disease characterized by incomplete vascularization of the peripheral retina in human patients. We initially assessed the vascular development in P5 iECKO/+ mice. The results exhibited no overt differences in vascular progression, vessel density, and vessel branchpoints between iECKO/+ mice and control littermates (*Figure 2—figure supplement 3*). This feature resembles the vascular development characteristics observed in other heterozygous mice carrying FEVR pathogenic genes, such as *Lrp5* (*Xia et al., 2008*). To assess the role of *CAPSL* in the growth and patterning of retinal vasculature, we further employed retinal flat-mount analysis to visualize the retinal vessels at different postnatal ages in homozygous mice. As expected, the radial vascular growth, vessel density, and vascular branching were dramatically reduced in P5 iECKO retina relative to Ctrl mice (*Figure 2A, D–F*). The number of retinal vascular tip cells, which leads EC

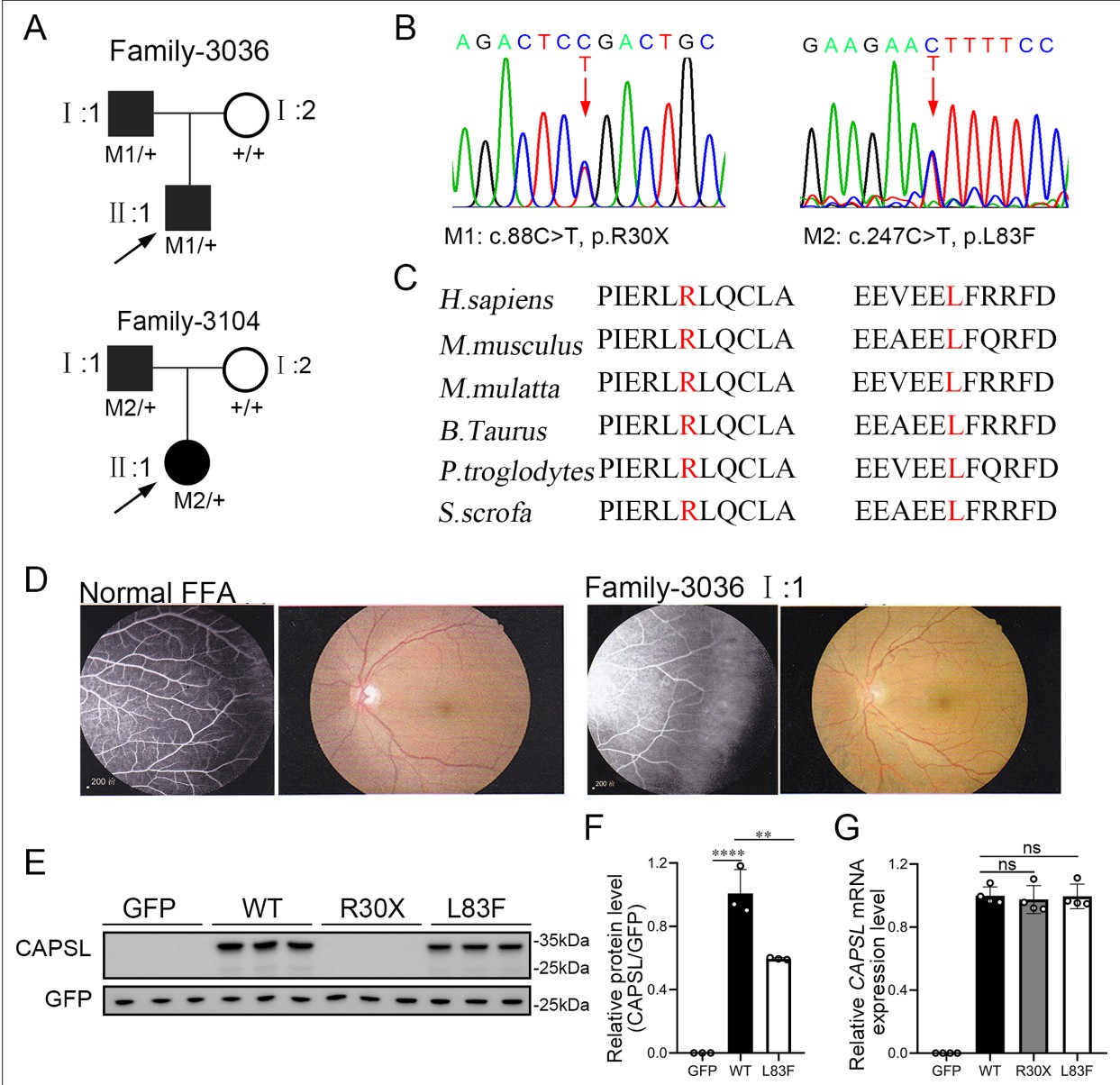

**Figure 1.** *CAPSL* point mutations in two families with familial exudative vitreoretinopathy (FEVR). (**A, B**) FEVR pedigrees (patients are denoted with black symbols) and Sanger sequencing of two heterozygous mutations identified in two families. Black arrows indicate the proband of each family and red arrows indicate the changed nucleotides. (**C**) Alignment of amino sequences surrounding the *CAPSL* variants in different species and all mutated sites are highly conserved. Altered amino acid residues are highlighted in red. (**D**) Fundus fluorescein angiography (FFA) (left panel) and fundus photography (right panel) of a normal individual and FEVR-affected patient (I:1) in family 3036. Western blot (**E, F**) and quantitative real time polymerase chain reaction (RT-qPCR) analysis (**G**) of *CAPSL* expression of WT and mutant plasmids. An empty vector with GFP tag was used as a negative control. GFP was used as an internal reference. Error bars indicate the standard deviation (SD). **p < 0.01, ****p < 0.0001, ns: no significance, by Student's *t* test (*n* = 3).

The online version of this article includes the following source data and figure supplement(s) for figure 1:

**Source data 1.** Uncropped and labeled gels for *Figure 1*.

**Source data 2.** Raw unedited gels for *Figure 1*.

**Figure supplement 1.** Subcellular localization of CAPSL variant proteins.

**Figure supplement 2.** Bioinformatic prediction of an impact on splicing of the pathogenic variant c.88C>T in *CAPSL*.

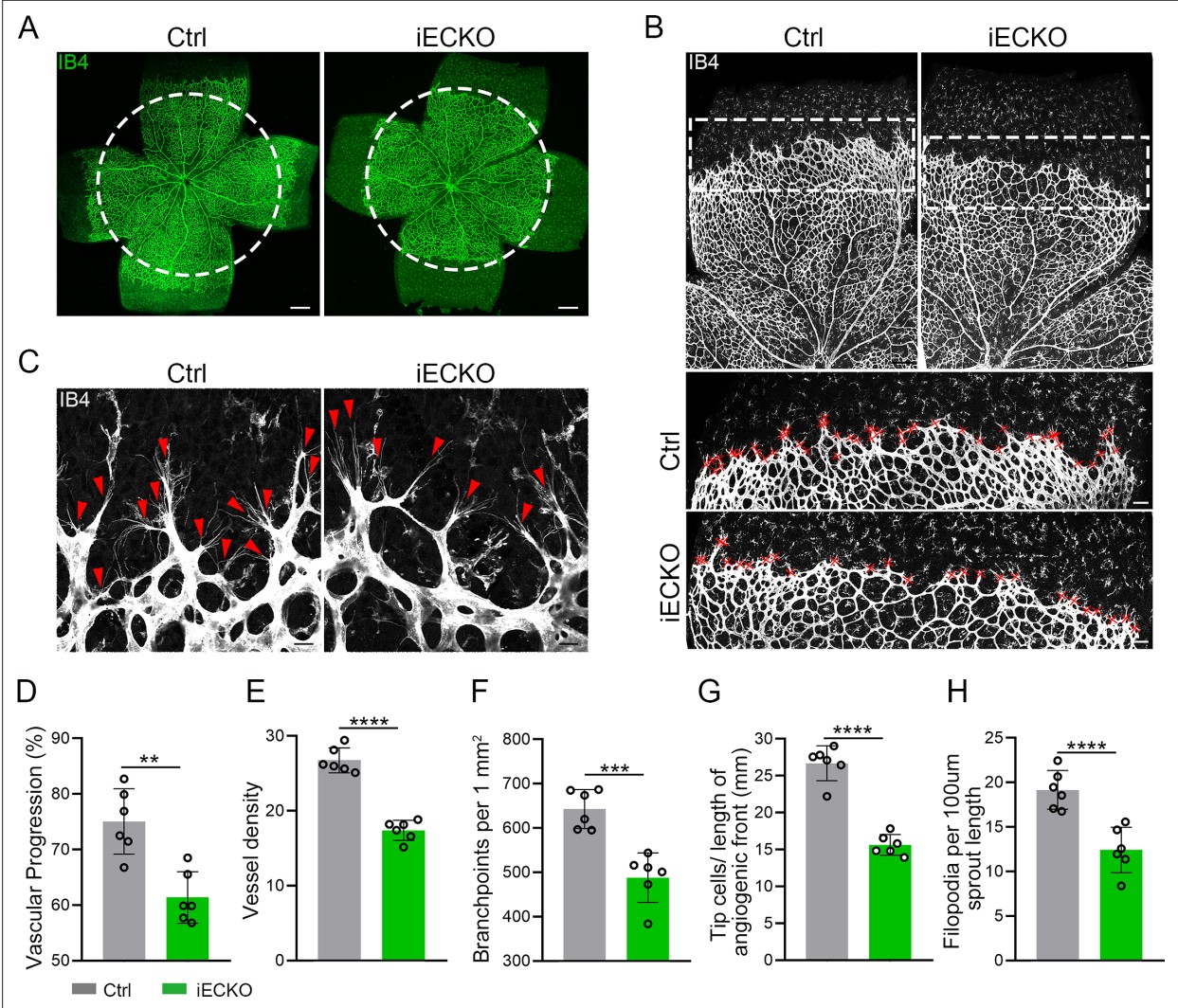

**Figure 2.** Endothelial cell (EC)-specific inactivation of *Capsl* impairs retina angiogenesis. (**A**) Flat-mounted retinas obtained from P5 Ctrl and littermate iECKO mice were stained with Isolectin-B4 (IB4) to visualize blood vessel. Dashed circle mark the edge of the developing retina vessel in iECKO mice. Scale bar: 250 µm. (**B**) Low magnification images (top panels) and high magnification images (bottom panels) in boxed areas of IB4-stained angiogenic front of Ctrl and iECKO mice, respectively. Red cross mark tip cells at the angiogenic growth front. Scale bar: 100 µm (top panels), 25 µm (bottom panels). (**C**) High magnification images of filopodia-extending cells at the edge of retinal angiogenic growth front from Ctrl and iECKO mice. Red arrowheads indicate the sprouts at the angiogenic growth front. Scale bar: 50 µm. (**D–H**) Quantification of retinal vascular development parameters, including vascular progression, vessel density, branchpoints, number of tip cells, and number of filopodia. Error bars indicate the standard deviation (SD). **$p < 0.01$, ***$p < 0.001$, ****$p < 0.0001$, by Student's *t* test ($n = 6$).

The online version of this article includes the following source data and figure supplement(s) for figure 2:

**Figure supplement 1.** Construction of endothelial cell (EC)-specific *Capsl*-knockout mice model and CAPSL knockdown human retinal microvascular endothelial cells (HRECs) model.

**Figure supplement 1—source data 1.** Uncropped and labeled gels for *Figure 2—figure supplement 1*.

**Figure supplement 1—source data 2.** Raw unedited gels for *Figure 2—figure supplement 1*.

**Figure supplement 2.** Experimental mice do not carry confounding *rd1* and *rd8* mutations.

**Figure supplement 3.** *Capsl* heterozygous mice exhibited no significant defect in angiogenesis.

migration toward high vascular endothelial growth factor (VEGF) by extending filopodia at the angiogenic front (*Rattner et al., 2019*), was considerably decreased and sparsely distributed in iECKO retinae, compared to that of control littermates (*Figure 2B, C, G, H*).

The onset of mouse retinal vascular development occurs postnatally at P0, coinciding with the regression of the hyaloid vasculature (*Saint-Geniez and D'amore, 2004*; *Ito and Yoshioka, 1999*).

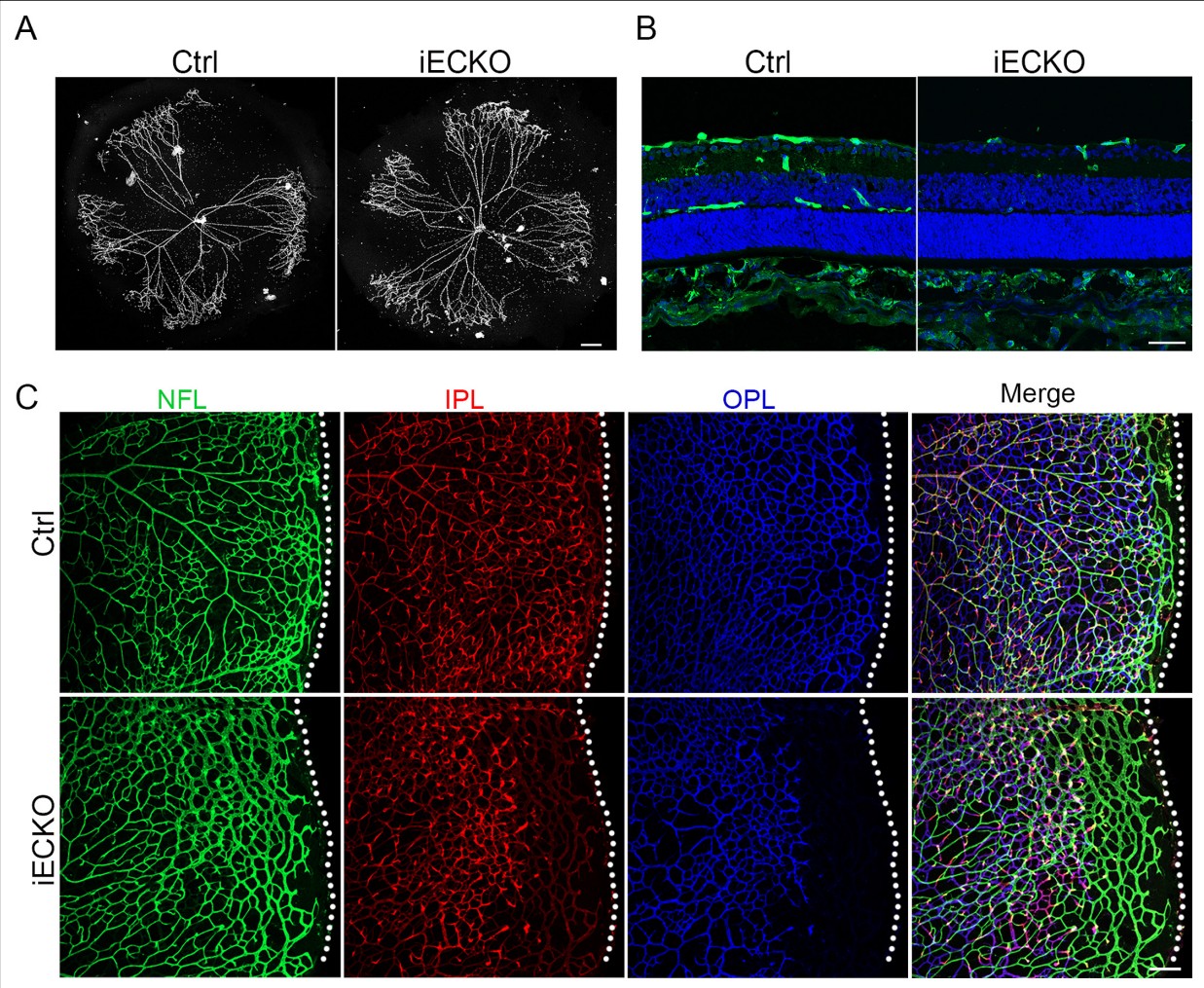

**Figure 3.** Loss of *Capsl* results in delayed hyaloid regression and deep retinal blood vessel growth. (**A**) Hyaloid vessels stained with DAPI in the eyes of Ctrl and iECKO mice at P10. Scale bar: 250 µm. (**B**) Retina sections from P10 Ctrl and iECKO mice were co-stained with Isolectin-B4 (IB4) (greed) and DAPI (blue). Scale bar: 100 µm. (**C**) Flat-mounted retains stained with IB4 at P14 Ctrl and iECKO mice. Optical sections of z-stacked confocal images were divided to represent the nerve fiber layer (NFL), inner plexiform layer (IPL), and outer plexiform layer (OPL). Dashed lines mark the edge of the developing retina. Scale bar: 100 µm.

The online version of this article includes the following figure supplement(s) for figure 3:

**Figure supplement 1.** Capsl deletion does not affect vessel maturation.

Flat-mounted retinas stained with Isolectin-B4 (IB4) at P21 Ctrl and iECKO. Optical sections of z-stacked confocal images were divided to represent the nerve fiber layer (NFL), inner plexiform layer (IPL), and outer plexiform layer (OPL). Scale bar: 250 µm (left panel) and 100 µm (right panel).

Similar to the previously reported hyaloid phenotypes observed in the FEVR-associated mouse models (*Junge et al., 2009*; *Chen et al., 2012*), the hyaloid vasculature in iECKO at P10 displayed delayed regression in comparison to the Ctrl littermates (*Figure 3A*). Following the expansion of the superficial vessel plexus at around P8, retinal vessels extend perpendicularly into the deep retina, forming secondary and deep third capillary layers (*Fruttiger, 2007*). To investigate whether the depletion of *Capsl* hinders deep capillary vascular development, we conducted immunostaining on P10 retinal sections. The results revealed a delayed development of deep retinal vessels in iECKO mouse retinae compared to that of Ctrl (*Figure 3B*). In addition, immunostaining of P14 flat-mount retina confirmed an incomplete architecture of the deep vascular layer in iECKO mouse (*Figure 3C*). By P21, EC growth was closely resembling control retinas in iECKO mice when angiogenesis almost complete (*Figure 3— figure supplement 1*).

## CAPSL controls vascular proliferation and migration

The development of retinal vasculature is a complex process encompassing not only the coordinated behavior of ECs but also the intricate interactions among different cell types and cytokines within the retina (*Selvam et al., 2018*). Astrocytes form a mesh as a template prior to vascular growth and guide angiogenic expansion by secreting gradient distributed VEGFA (*Dorrell et al., 2002*; *Gerhardt et al., 2003*; *Stenzel et al., 2011*). Abnormal activation of astrocytes and subsequent inflammatory responses have been observed in multiple FEVR mouse models (*Junge et al., 2009*; *Zhu et al., 2021*; *Fruttiger, 2007*; *Luhmann et al., 2005*). To explore whether the gradient and expression level of VEGFA were affected by the depletion of *Capsl*, we conducted immunostaining of VEGFA on the retina. However, the results showed no alteration of VEGFA (*Figure 4—figure supplement 1A*). Meanwhile, the activation state and the pattern of the vascular template for EC outgrowth formed by astrocytes remained normal, indicated by glial fibrillary acidic protein (GFAP) (*Figure 4—figure supplement 1B*). Pericyte recruitment is necessary for vessel stability and maturation (*Hellström et al., 2001*). The immunostaining of Desmin or NG2 staining exhibited no difference between Ctrl and iECKO mice (*Figure 4—figure supplement 1C*). To assess the integrity of the retinal vascular barrier, we performed the intraperitoneal injection of 5- (and-6)-tetramethylrhodamine biocytin (biocytin-TMR, 869 Da), a small-molecular-weight fluorescent tracerm (*Knowland et al., 2014*). The results showed no notable leakage in the iECKO retina, suggesting the intact blood barrier despite the depletion of ECs-*Capsl* (*Figure 4—figure supplement 1D*). The extracellular matrix (ECM) coordinates the behavior of ECs by regulating cell–cell communication processes (*Liu and Senger, 2004*). The deficiency and abnormal accumulation of ECM can both disrupt the adhesion, proliferation, and migration of the ECs, leading to defects in blood vessel architecture (*Fujiwara et al., 2004*; *Fischer et al., 2009*; *Santos-Oliveira et al., 2015*). We thus analyzed the ECM accumulation using fibronectin 1, and the result showed normal deposition of ECM (*Figure 4—figure supplement 1E*). In conclusion, EC-specific knockout of *Capsl* may mainly lead to defects in retinal vascular ECs rather than other vascular-associated cells.

Considering the compromised vascular progression and decreased vessel density in iECKO mouse retinae, we first performed the EdU assay on retina flat mounts co-stained with ETS transcription factor ERG (endothelial cell nuclear marker) to assess the proliferation of ECs. The results showed a strongly reduced proliferation of *Capsl*-defective ECs compared to that of Ctrl (*Figure 4A*). As previously reported, the regressed retinal vessels exhibited sleeve-like positivity for basement matrix component Collagen IV and negativity for Isolectin IB4 (*Birdsey et al., 2015*). To compare the vascular regression between iECKO and Ctrl retinae, we conducted co-staining of the Collagen IV and IB4. The results revealed that depletion of *Capsl* leads to an increased vascular regression, evidenced by a significant increase in Collagen IV basal membrane sleeves lacking IB4 both in the capillary plexus and angiogenic front (*Figure 4B*).

Additionally, considering the pivotal role of tip cells in the direction of vascular development, we then asked whether depletion of *Capsl* impairs the polarity of tip cells by analyzing the morphology of cell nuclei at the angiogenic front. This analysis was based on the observation that the nuclei of actively migrating tip cells are elliptical and exhibit increased sphericity (decreased ellipticity) in static tip cells (*Coxam et al., 2014*; *Kim et al., 2019*). In Ctrl mice retinae, the nuclei of tip ECs were predominantly elliptical, pointing toward the avascular area (*Figure 4C*). Conversely, in the retinae of iECKO mice, the nuclei of tip ECs appeared more spherical and did not orient toward the avascular area (*Figure 4C*). These findings suggest that CAPSL is critical for EC proliferation, vascular regression, and cell polarity.

## Endothelial CAPSL is required for cell polarity and filopodia/lamellipodia formation

To investigate the functional mechanism and signaling cascade by which CAPSL regulates EC behaviors, we performed in vitro experiments on primary human retinal microvascular endothelial cells (HRECs). Stable *CAPSL* knockdown HRECs were generated using the lentivirus-mediated shRNA system. RT-qPCR analysis showed that the mRNA level of *CAPSL* was significantly decreased (*Figure 2—figure supplement 1E*). In line with the overall reduced vascular coverage in iECKO mice retina, *CAPSL*-knockdown HRECs (sh*CAPSL*-ECs) exhibited considerably impaired tube formation in vitro (*Figure 5A*). EdU labeling also revealed decreased cell proliferation of sh*CAPSL*-ECs compared

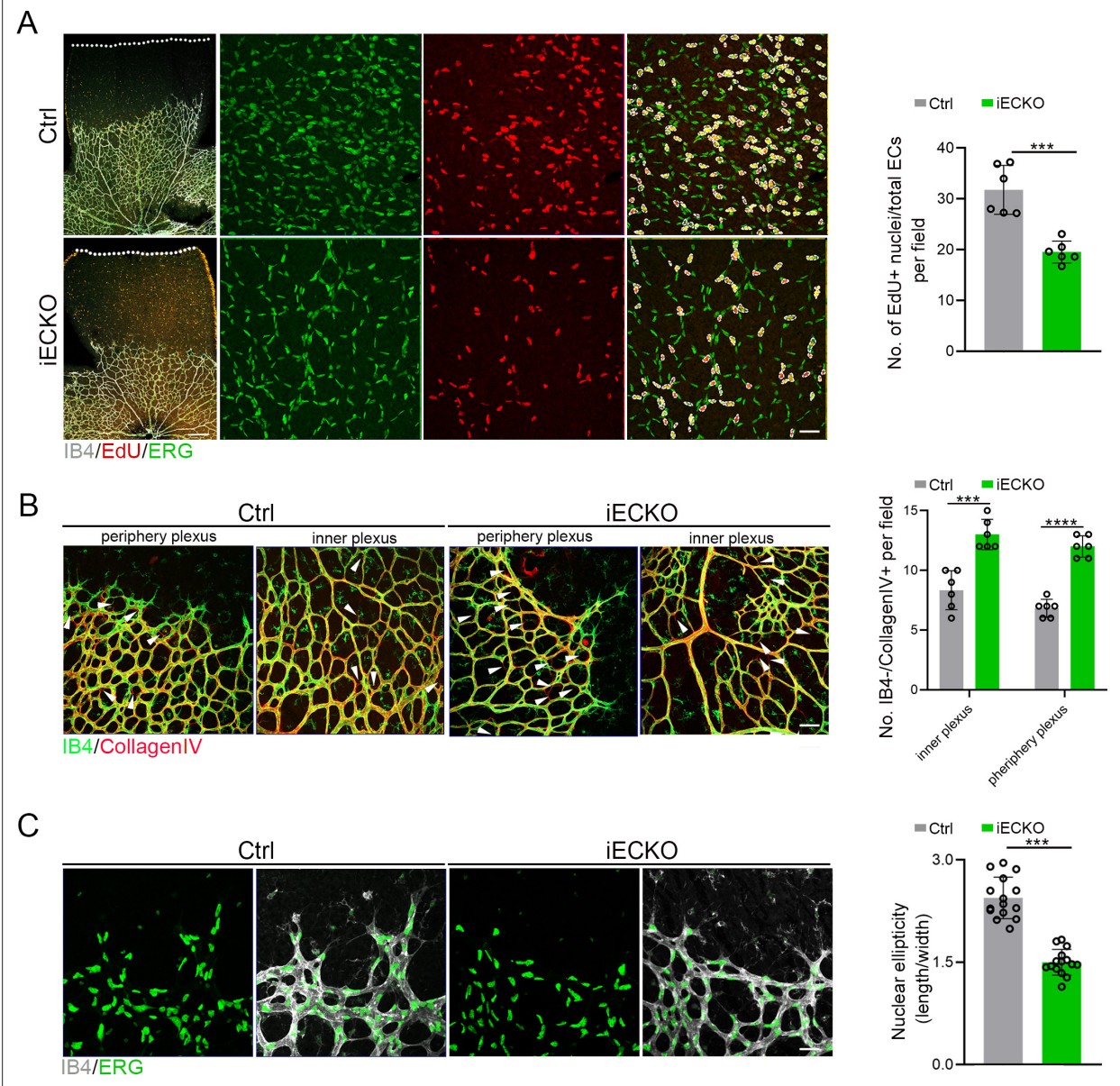

**Figure 4.** Deletion of *Capsl* impairs endothelial cell (EC) proliferation and migration. (**A**) Retina EC proliferation of Ctrl and iECKO mice at the vitreal surface was measured with EdU and ERG labeling at P5. Images captured at higher magnification are shown at right. Dashed lines mark the edge of the developing retina, and dashed circles represent both EdU+ and ERG+ cells. EC proliferation ability was measured by the ration of EdU+ and ERG+ cells per vascular area. Scale bar: 200 and 50 µm (enlarged insets). Error bars indicate the standard deviation (SD). ***p < 0.001, by Student's *t* test (*n* = 6). (**B**) Representative images of retinal vessels at the periphery plexus and inner plexus of Ctrl and iECKO mice at P5 co-stained with Isolectin-B4 (IB4) (green) and Collagen IV (red). Arrowhead point to empty Collagen IV sleeves. And quantification of ratio of Collagen IV-positive vessel segments to IB4 labeling-negative vessel segments. Scale bar: 50 µm. Error bars indicate the standard deviation (SD). ***p < 0.001, ****p < 0.0001, by Student's *t* test (*n* = 6). (**C**) Magnified images of IB4+ vessels and ERG+ nuclei of ECs at angiogenic front of Ctrl and iECKO mice at P5. Quantification of tip cell nuclear ellipticity at the angiogenic growth front. Scale bar: 50 µm. Error bars indicate the standard deviation (SD). ***p < 0.001, by Student's *t* test (*n* = 14).

The online version of this article includes the following figure supplement(s) for figure 4:

**Figure supplement 1.** Normal state of other cell types and extracellular matrix (ECM) deposition in *Capsl*-knockout mice retina.

to that of shCtrl-ECs (*Figure 5B*). We further conducted flow cytometry to determine whether cell cycle arrest occurred upon depletion of *CAPSL* in HRECs. The redistribution of cell cycle phases indicated that the cell cycle was arrested during the G0/G1 phase in the sh*CAPSL*-ECs (*Figure 5C, D*). To further investigate the role of CAPSL in EC migration, we performed the scratch-wound assay in confluent sh*CAPSL*-ECs and shCtrl-ECs, as previously reported (*Kim et al., 2019*; *Carvalho et al.,*

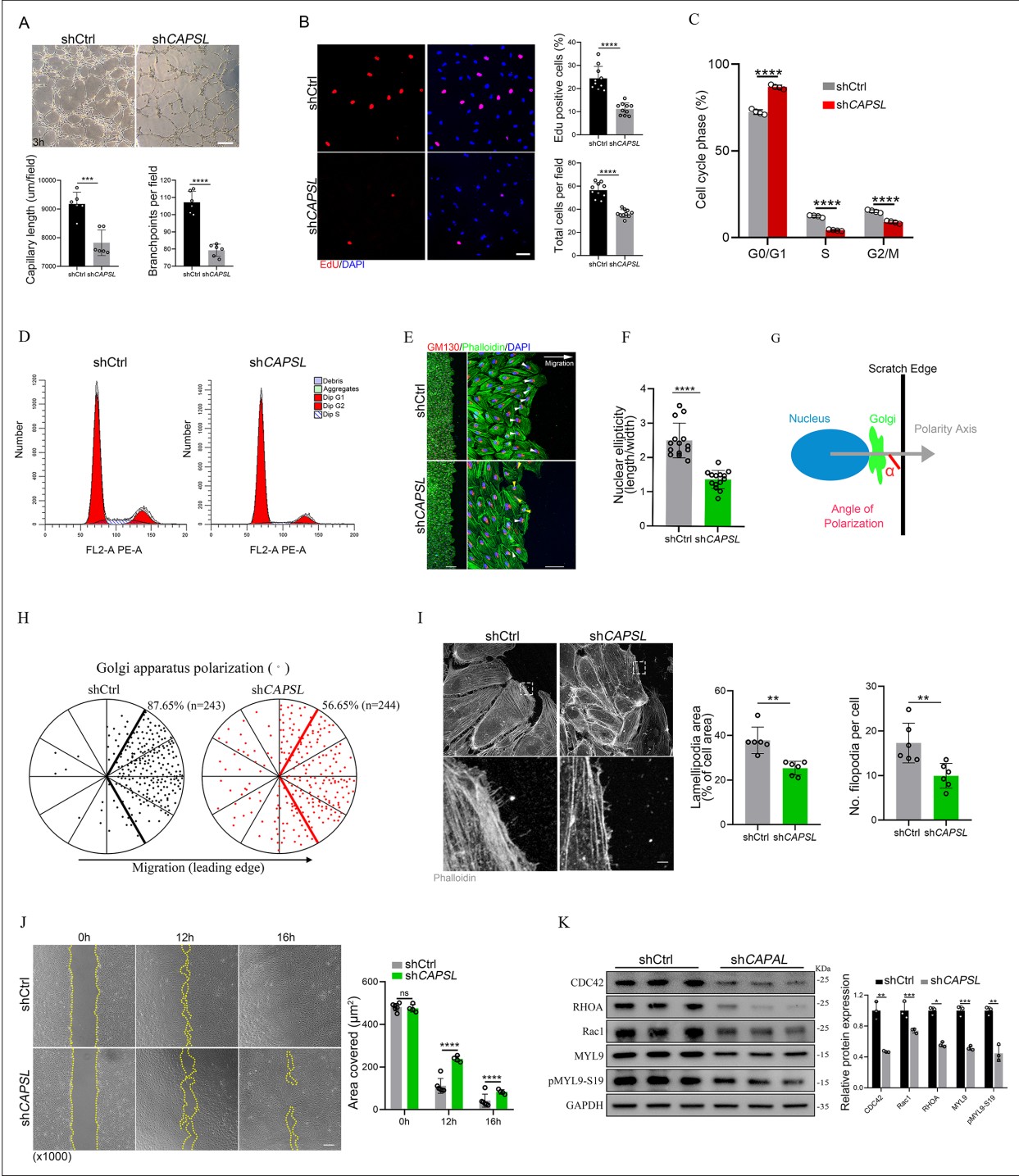

**Figure 5.** Depletion of *CAPSL* in human retinal microvascular endothelial cells (HRECs) compromises in vitro endothelial cell (EC) proliferation and migration. (**A**) Representative images of in vitro tube formation after transfection of HRECs with shRNA. Scale bar: 200 μm. Error bars indicate the standard deviation (SD). ***p < 0.001, ****p < 0.0001, by Student's *t* test (*n* = 6). (**B**) Incorporation of EdU in shRNA transfected HRECs. Representative confocal images and quantification of proliferating HRECs both in number per field and proportion of EdU-positive cells. Scale bar: 50 μm. Error bars indicate the standard deviation (SD). ****p < 0.0001, by Student's *t* test (*n* = 10). (**C, D**) Cell cycle analysis of shCtrl-ECs and sh*CAPSL*-ECs by flow cytometry. Error bars indicate the standard deviation (SD). ****p < 0.0001, by Student's *t* test (*n* = 4). (**E**) Representative images of phalloidin actin cytoskeleton (green) and GM130 (red) showing polarity angles of shCtrl-ECs and sh*CAPSL*-ECs at the edge of scratch wound. The arrow points toward the wound. Colored arrowheads represent different migration state. Scale bar: 200 μm (left panel), 50 μm (right panel). (**F**) Quantification of nuclear ellipticity of HRECs at the margin of wound scratch. Error bars indicate the standard deviation (SD). ****p < 0.0001, by Student's *t* test (*n* = 14). (**G**) Schematic pictures showing the define of polarity axis of each cell. Polarity axis was measured with the angle (α) between the scratch edge

*Figure 5 continued on next page*

*Figure 5 continued*

and the vector drawn from the center of nucleus to the center of the Golgi apparatus. (**H**) Polar plots showing Golgi apparatus polarization. The bold lines represent 120° region centered on the vector, which is perpendicular to the wound scratch. The dots represent the angle (α) of each cell and the numbers indicate the frequency of dots within the 120° region of the bold line of shCtrl-ECs (*n* = 243) and sh*CAPSL*-ECs (*n* = 244). (**I**) Images of phalloidin-stained actin cytoskeleton and comparisons of indicated parameters in shCtrl-ECs and sh*CAPSL*-ECs at the edge of scratch wound. The dashed boxed region is shown at higher magnification at the bottom panel. Scale bar: 50 μm (top panels), 25 μm (bottom panels). Error bars indicate the standard deviation (SD). **p < 0.01, by Student's *t* test (*n* = 6). (**J**) Representative images of wound scratch assay at 0, 12, and 16 hr after wound was made. And the quantification of covered area at different time point. The dashed line indicates the gap of the wound after wound scratch at different time point. Scale bar: 200 μm. Error bars indicate the standard deviation (SD). ****p < 0.0001, ns : no significance, by Student's *t* test (*n* = 6). (**K**) Immunoblot and quantification analysis of expression of small GTPase proteins and a key regulator of contractile force MYL9 in shCtrl-ECs and sh*CAPSL*-ECs. Error bars indicate the standard deviation (SD). *p < 0.05, **p < 0.01, ***p < 0.001, by Student's *t* test (*n* = 3).

The online version of this article includes the following source data for figure 5:

**Source data 1.** Uncropped and labeled gels for *Figure 5*.

**Source data 2.** Raw unedited gels for *Figure 5*.

*2019*). Nine hours after scratching, ECs were co-stained with GM130 and phalloidin to visualize the Golgi apparatus and the cytoskeleton. The results indicated that the nuclei of sh*CAPSL*-ECs were more spherical than those of shCtrl-ECs (*Figure 5E, F*). Given that the axial polarity can reflect the direction and migration of ECs (*Franco et al., 2015*; *Kwon et al., 2016*), we further measured the cell polarity (nucleus-to-Golgi polarity axis) of ECs at the wound edge (*Kim et al., 2019*; *Carvalho et al., 2019*). As a result, the Golgi apparatus of shCtrl-ECs was predominantly positioned around the wound edge, while that of sh*CAPSL*-ECs showed more random positioning (*Potente et al., 2011*; *Figure 5H*).

At the angiogenic front, the dynamic behaviors of tip cells largely rely on the rearrangement and organization of the actin cytoskeleton (*De Smet et al., 2009*). Tip cells extend actin-driven filopodia and lamellipodia to probe chemotactic guidance cues, providing direction and migration for the developing vascular network (*Kurz et al., 1996*; *Marin-Padilla, 1985*; *Ruhrberg et al., 2002*). Notably, sh*CAPSL*-ECs exhibited fewer filopodia, smaller lamellipodia, and significantly impaired wound closure compared to shCtrl-ECs at the wound scratch edge (*Figure 5I*). Also, as expected, sh*CAPSL*-ECs exhibited significantly impaired wound closure at the same time point after scratching relative to shCtrl-ECs (*Figure 5J*). Small Rho GTPases, including CDC42 and Rac, are essential for lamellipodial extension and cell migration (*Fraccaroli et al., 2015*; *Hall, 2005*). Western blot analysis uncovered markedly attenuated expression of CDC42, RHOA, and Rac1 (*Figure 5K*). Moreover, both MYL9 and phosphorylated MYL9 at Ser19, which are crucial regulators of contractile force, were also drastically reduced in sh*CAPSL*-ECs compared to those of shCtrl-ECs (*Figure 5K*). These results indicated that CAPSL regulates EC polarity, filipodia/lamellipodia formation, and migration by modulating the actomyosin cytoskeleton.

## CAPSL regulates EC proliferation through the MYC signaling axis

Given the fact that FEVR is predominantly characterized by the compromised Norrin/β-catenin signaling pathway as its main causative factor (*Chen et al., 1993*; *Junge et al., 2009*; *Zhu et al., 2021*; *Panagiotou et al., 2017*), we first asked whether loss of CAPSL function causes FEVR through Norrin/β-catenin signaling. TopFlash reporter gene system was used to determine the activity of the Norrin/β-catenin signaling pathway. Knocking down *CAPSL* expression in human embryonic kidney 293 (HEK293) SuperTopFlash (STF) cells led to similar luciferase activity, compared to cells transfected with shCtrl (*Figure 6—figure supplement 1A*). Furthermore, transfection of *shRNA*-resistant CAPSL plasmid (WT, R30X, or L83F) in *CAPSL*-knockdown 293STF cells resulted in no significant difference in luciferase activity (*Figure 6—figure supplement 1B*). These results indicated that *CAPSL* was not a major player in Norrin/β-catenin signaling to regulate retinal vascular development.

We next performed unbiased transcriptomic and proteomic analyses on the control and *CAPSL*-knockdown HRECs to explore the underlying mechanism by which *CAPSL* regulates EC proliferation and migration. RNA sequencing (RNA-seq) identified 1961 up-regulated genes and 2205 down-regulated genes in sh*CAPSL*-ECs compared to shCtrl-ECs (logFC >0.58, p-value <0.05), 1342 up-regulated genes and 1218 down-regulated genes in ECs overexpressing *CAPSL* (LentiOE-ECs) compared to Ctrl-ECs (*Figure 6A*). Gene set enrichment analysis (GSEA) on transcriptomic data

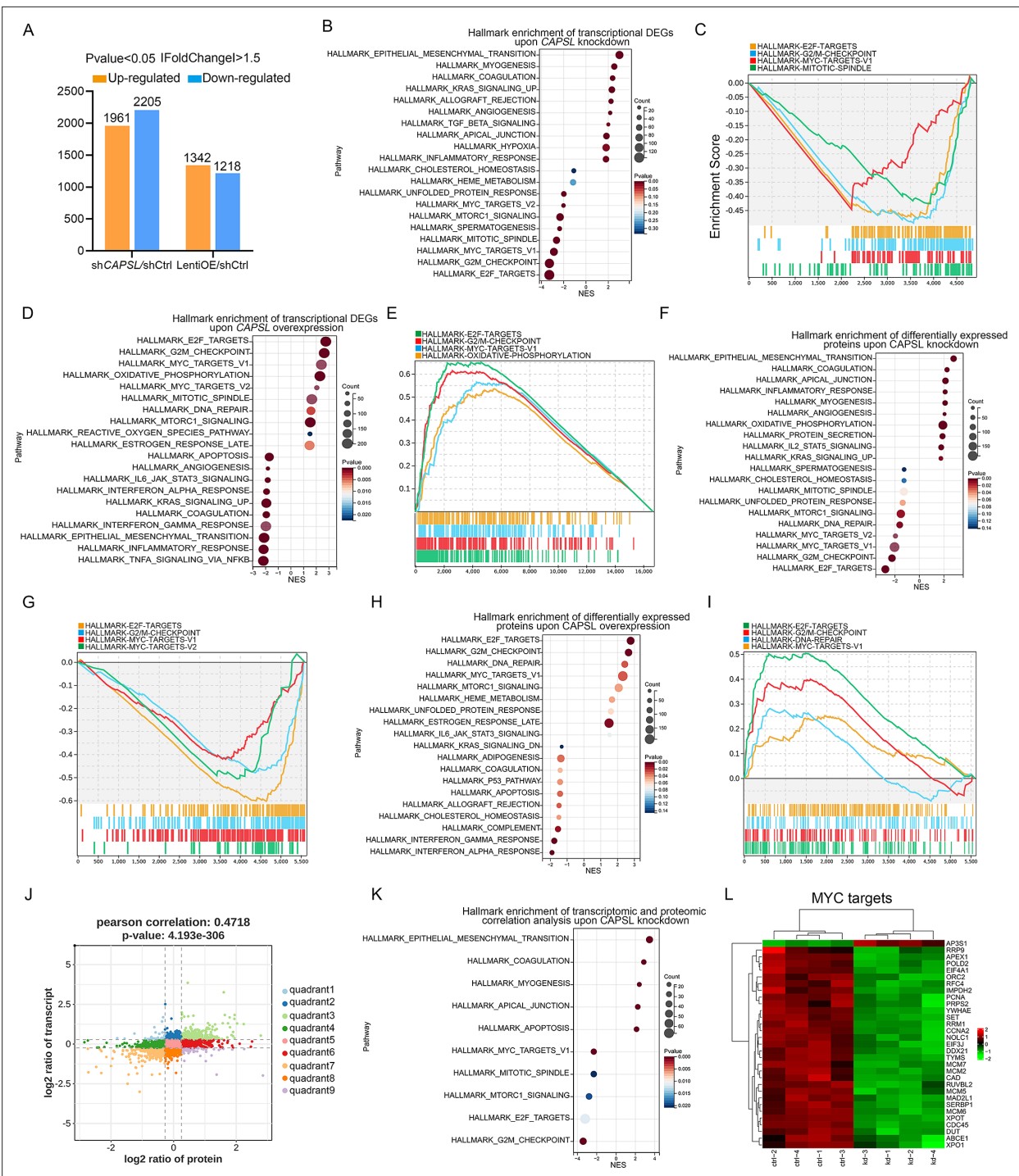

**Figure 6.** *CAPSL* suppresses MYC signaling axis. (**A**) Differential gene expression information of sh*CAPSL*-ECs versus shCtrl-ECs group and LentiOE-ECs versus shCtrl-ECs group. (**B–E**) Gene set enrichment analysis (GSEA) on the RNA sequencing data of human retinal microvascular endothelial cells (HRECs). Top 10 ranked up- or down-regulated signaling axis was listed (**B**) and top 4 down-regulated gene sets were listed (**C**) in comparison of shCtrl-ECs versus sh*CAPSL*-ECs. Top 10 ranked up- or down-regulated signaling axis was listed (**D**) and top 4 up-regulated gene sets were listed (**E**) in comparison of LentiOE-ECs versus shCtrl-ECs. (**F–I**) GSEA on the proteomic profiling data of HRECs. Top 10 ranked up- or down-regulated signaling axis was listed (**F**) and top 4 down-regulated gene sets were listed (**G**) in comparison of shCtrl-ECs versus sh*CAPSL*-ECs. Top 10 ranked up- or down-regulated signaling axis was listed (**H**) and top 4 up-regulated gene sets were listed (**I**) in comparison of LentiOE-ECs versus shCtrl-ECs. (**J**) Correlated RNAs and proteins enriched in nine quadrants of shCtrl-ECs versus sh*CAPSL*-ECs. (**K**) GSEA on the genes/proteins in quadrants 3 and 7, and top 5 ranked up- or down-regulated signaling axis was listed. (**L**) Clustered heatmap of the expression fold changes of several MYC signature genes of in both RNA profiling and proteomic profiling of shCtrl-ECs and sh*CAPSL*-ECs.

*Figure 6 continued on next page*

*Figure 6 continued*

The online version of this article includes the following source data and figure supplement(s) for figure 6:

**Figure supplement 1.** CAPSL is not involve in Norrin/β-catenin signaling pathway.

**Figure supplement 2.** *Capsl* depletion resulted in downregulation of MYC targets.

**Figure supplement 2—source data 1.** Uncropped and labeled gels for *Figure 6—figure supplement 2*.

**Figure supplement 2—source data 2.** Raw unedited gels for *Figure 6—figure supplement 2*.

showed multiple down-regulated signaling axes upon the depletion of *CAPSL*, including E2F targets (NES = −3.2944438, p-value = 0), G2/M checkpoints (NES = −3.2851038, p-value = 0), MYC targets (NES = −2.8795638, p-value = 0), and mitotic spindle (NES = −2.646523, p-value = 0) (*Figure 6B, C*). In line with this, overexpression of CAPSL in HRECs also caused a significant increase in E2F targets (NES = 2.7765026, p-value = 0), G2/M checkpoints (NES = 2.6528199, p-value = 0), and MYC targets (NES = 2.6528199, p-value = 0) (*Figure 6D, E*). Meanwhile, proteomic profiling showed similar expression patterns with that of transcriptomic profiling upon knockdown or overexpression of *CAPSL* (*Figure 6F–I*). Furthermore, we applied correlation analysis on the transcriptomic and proteomic data, which divided the genes into nine quadrants (*Figure 6J*). GSEA analysis was performed on genes in the third and seventh quadrants, which were regulated at the transcriptional level (*Figure 6K* and *Supplementary file 2*). Interestingly, the down-regulated signaling pathways are highly consistent with those observed in GSEA analysis of both transcriptomic and proteomic analyses (*Figure 6K*). Given that the MYC was an established driver of cell growth and migration (*Lourenco et al., 2021*), and that the genes in the MYC signaling were down-regulated upon *CAPSL* depletion (*Figure 6L*), we speculated that CAPSL might play a pivotal role in MYC signaling in HRECs. RT-qPCR analysis was further performed to validate the downregulation fo MYC target genes in the lung tissue of Ctrl and iECKO mice (*Figure 6—figure supplement 2*).

We next incorporated an analysis of transcriptome profiling of human umbilical vein endothelial cells (HUVECs) infected with lentiviruses encoding control- or *MYC*-targeting genomic RNAs (gRNAs; GSE161815) (*Andrade et al., 2021*). Intriguingly, GSEA analysis revealed that ablation of cMYC in HUVECs led to compromised E2F targets, MYC targets, and G2/M checkpoint signaling activity (*Figure 7A, B*), largely consistent with those observed in the absence of *CAPSL*. This also suggests that cMYC might play a role in the upstream regulation of E2F targets and G2/M checkpoint signaling. Venn diagram analysis of down-regulated genes (LogFC <−0.585, p < 0.05) in *CAPSL*-depleted HRECs and *MYC*-depleted HUVECs revealed a large proportion of shared genes, indicative of similar transcriptional regulatory patterns between CAPSL and MYC (*Figure 7C* and *Supplementary file 3*). Additionally, these genes are predominently enriched in the MYC targets signaling (*Figure 7D–F*).

We, therefore, asked whether CAPSL regulates MYC expression. Notably, the depletion of *CAPSL* significantly decreased the protein level of MYC, rather than the mRNA level in HRECs (*Figure 7G*). Subsequently, western blot analysis revealed that expression levels of MYC target genes, such as MCM family, E2F1, CyclinD family, and PCNA, were decreased in the absence of *CAPSL* in HRECs (*Figure 7H, I*). Taken together, these findings suggest MYC as a potential functional regulator downstream of CAPSL.

## Discussion

Over the last decades, major progress has been made in understanding the molecular mechanisms underlying FEVR, and 17 FEVR-causing genes have been identified (*Chen et al., 1993*; *Robitaille et al., 2002*; *Gong et al., 2001*; *Jiao et al., 2004*; *Li et al., 2022*; *Junge et al., 2009*; *Nikopoulos et al., 2010*; *Zhu et al., 2021*; *Panagiotou et al., 2017*; *Liu et al., 2024*; *He et al., 2023*; *Yang et al., 2022*; *Collin et al., 2013*; *Robitaille et al., 2014*; *Khan et al., 2012*; *Downey et al., 2001*; *Park et al., 2019*; *Zhang et al., 2020*; *Zhang et al., 2021*; *Asano et al., 2021*; *Xu et al., 2023*; *Li et al., 2023*). However, the pathogenic genes and mechanisms underlying approximately 50% of clinical cases remain elusive (*He et al., 2023*; *Yang et al., 2022*). In this study, we report a new FEVR candidate gene *CAPSL* and uncover the pivotal roles of CAPSL in retina vascular development. The EC-specific *Capsl*-knockout mouse model exhibited FEVR phenotypes, including delayed vascular progression, retarded hyaloid vessel regression, and decreased vessel density. Deficiency of CAPSL in ECs resulted

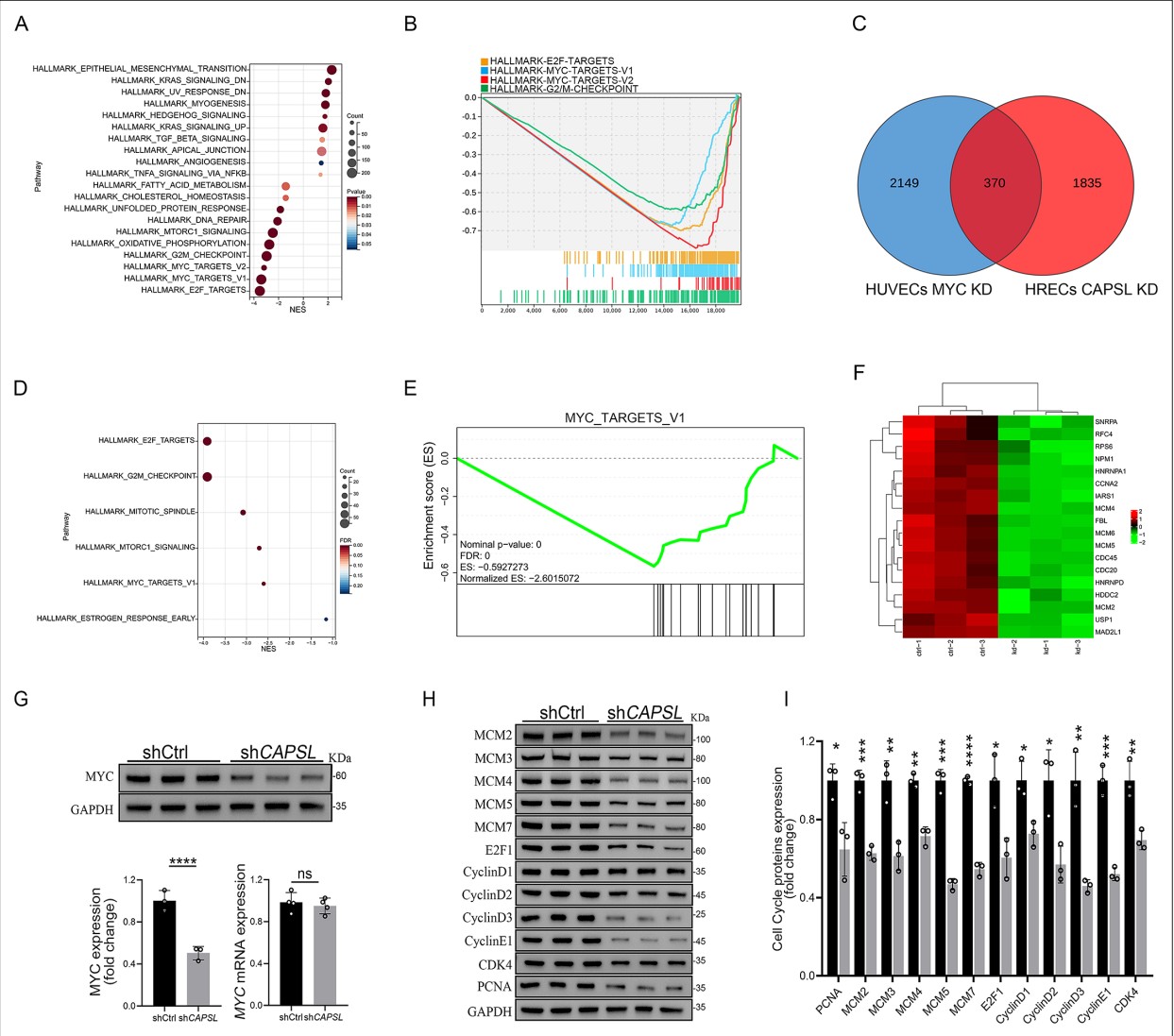

**Figure 7.** Loss of *CAPSL* led to similar transcriptional regulatory patterns to the loss of *MYC* in human umbilical vein endothelial cells (HUVECs). (**A, B**) Gene set enrichment analysis (GSEA) on the RNA sequencing data in comparison of shCtrl-HUVECs versus sh*MYC*-HUVECs. Top 10 ranked up- or down-regulated signaling axis was listed (**A**) and top 4 down-regulated gene sets were listed (**B**). (**C**) Venn diagram analysis of down-regulated genes in *CAPSL*-depleted human retinal microvascular endothelial cells (HRECs) and c*MYC*-depleted HUVECs. (**D, E**) GSEA on the shared down-regulated genes in *CAPSL*-depleted HRECs and *MYC*-depleted HUVECs. (**F**) Heatmap of MYC signature genes of shared genes based on Venn analysis. (**G**) MYC expression level in shCtrl-ECs and sh*CAPSL*-ECs was quantified by western blot and RT-qPCR. Error bars indicate the standard deviation (SD). ****p < 0.0001, ns: no significance by Student's *t* test (*n* = 3). (**H**) Western blot analysis of expression of MYC targets. (**I**) Quantification analysis of MYC targets. Error bars indicate the standard deviation (SD). *p < 0.05, **p < 0.01, ***p < 0.001, ****p < 0.0001, by Student's *t* test (*n* = 3).

The online version of this article includes the following source data for figure 7:

**Source data 1.** Uncropped and labeled gels for *Figure 7.*

**Source data 2.** Raw unedited gels for *Figure 7*.

in compromised cell proliferation and defective EC migration, providing insights into the regulatory roles of CAPSL in retina vascularization (*Figures 2–5*).

During retinal development, angiogenesis comprises various processes such as EC specification, adhesion, proliferation, migration, and pruning (*Fruttiger, 2007*). The regulation of angiogenesis entails the control of EC behaviors and the interactions between ECs, ECM, and other types of retinal cells (*Franco et al., 2015*). In this study, we observed impaired radial expansion and vertical invasion, as well as the increased pruning of the retinal vascular ECs upon inactivating endothelial *Capsl* (*Figures 2 and 3*). However, neither the expression level nor the gradience of VEGFA was disturbed

in iECKO mice, which could be attributed to the intact blood barrier in the iECKO retina (*Figure 4—figure supplement 1*). In addition, the loss of *Capsl* resulted in the presence of more spherical tip cell nuclei, which were directed more randomly to the avascular area (*Figure 4*). These results provide evidence that CAPSL acts as a novel regulator for collective EC behavior.

In vitro functional studies demonstrated that depletion of *CAPSL* impaired tube formation, EC proliferation ability, and EC polarity (*Figure 5*). Moreover, the formation of filopodia and lamellipodia is also disturbed by the depletion of *CAPSL* (*Figure 5*). Using the unbiased transcriptomic and proteomic sequencing, bioinformatic analysis, and western blot assay, we demonstrated that the defects in *CAPSL* affect EC function by down-regulating the MYC signaling cascade (*Figures 6 and 7*). MYC has been shown to be a critical mediator of anabolic metabolism, cell growth, and migration (*Dang, 2013*; *Adhikary and Eilers, 2005*). Aberrant upregulation of the MYC signaling pathway is frequently observed in many types of cancers (*Dang, 2012*; *Stine et al., 2015*). Interestingly, the phenotypes of retinal vessels in iECKO mice resemble the vascular defects in *Myc^iEC-KO* mice, exhibiting FEVR-like phenotypes, such as impaired vascular progression, decreased branch points, and reduced EC proliferation (*de Alboran et al., 2001*; *Wilhelm et al., 2016*).

The Norrin/β-catenin signaling pathway is a subset of Wnt signaling pathway that is specifically activated during vascular development of the central nervous system (*He et al., 2023*; *Martowicz et al., 2019*). Although several signaling pathways have been implicated in the pathogenesis of FEVR, variants in crucial components of Norrin/β-catenin signaling account for most of the FEVR cases (*Junge et al., 2009*; *Zhu et al., 2021*; *Panagiotou et al., 2017*). It is worth mentioning that MYC is widely recognized as a downstream target of Wnt signaling, which is transcriptionally regulated during Wnt activation (*Green et al., 2024*; *Rennoll and Yochum, 2015*; *Cairo et al., 2012*). However, to our knowledge, there is a lack of literature reporting on the correlation between the MYC signaling pathway and FEVR. Interestingly, here we proved that the absence of CAPSL leads to a decrease in MYC protein level, which down-regulates MYC signaling without affecting Norrin/β-catenin signaling (*Figure 7* and *Figure 6—figure supplement 1*). Consequently, we propose MYC as a potential therapeutic target for FEVR, a hypothesis that warrants substantiation through subsequent investigations.

Additionally, the formation of filopodia and lamellipodia, which is reported to be intimately related to the actomyosin dynamics and the activity of small Rho GTPases such as CDC42 (*Barry et al., 2015*; *Zihni et al., 2016*), is significantly reduced in *CAPSL*-defective HRECs. In our study, we confirmed that the knockdown of *CAPSL* led to a significant reduction of CDC42, which might be the causative mechanism behind the disturbed formation of filopodia and lamellipodia (*Figure 5*).

Taken together, our study demonstrated that *CAPSL* is potentially a novel candidate gene associated with FEVR. We also demonstrated that variants in the *CAPSL* gene, independently of canonical Norrin/β-catenin signaling, cause FEVR through inactivating MYC signaling, expanding FEVR-involved signaling pathways.

## Materials and methods

### DNA sequencing of patients and controls

The study was approved by the Institutional Research Committees of Sichuan Provincial People's Hospital (approval number: NSFC2014-009). Informed consent was obtained from all participants and from guardians of minors involved in genetic testing. The patients were clinically diagnosed with exudative vitreoretinopathy. Genomic DNA was extracted from the peripheral blood of FEVR families and control individuals using a blood DNA extraction kit (QIAGEN, Germantown, Maryland, USA) according to standard protocol. WES was performed on DNA samples to identify candidate pathogenic genes for FEVR as previously described (*Zhu et al., 2021*). Family ID numbers 3036 and 3104 are not known to anyone outside the research group.

### Mouse model and genotyping

*Capsl^loxp/+* (named *Capsl^em1Zxj*) model was generated using the CRISPR/Cas9 nickase technique by Viewsolid Biotechology (Beijing, China) in C57BL/6J background. The gRNA sequence was as follows: Capsl-L gRNA: 5'-CTATCCCAA TTGTGCTCCTGG-3'; Capsl-R gRNA: 5'-TGGGACTCATGGTTCTAGAG G-3'. Pdgfb-iCreER (*Claxton et al., 2008*) transgenic mice on a mixed background of C57BL/6 and CBA was obtained from Dr. Marcus Fruttiger and backcrossed to background for 6 generations.

*Capsl*^loxp/+ mice were bred with Pdgfb-iCreER (*Claxton et al., 2008*) transgenic mice to generate *Capsl*^loxp/loxp, Pdgfb-iCreER mice. Genomic DNA samples extracted from mouse tails were genotyped using PCR to detect loxp-flanked *Capsl* and Pdgfb-iCreER. Primer sets used for *Capsl* loxp were: 5'-GGCAGGTAAGATGGTGTGTC-3' and 5'-TCTGTTTGTGGATCAATGTG-3'; and primer sets used for Pdgfb-iCreER were: 5'-GCCGCCGGGATCACTCTCG-3' and 5'-CCAGCCGCCGTCGCAACTC-3'. All mice were maintained as breeding colonies with standard rodent water and diet in a 12:12-hr dark cycle under specific pathogen-free conditions, and both males and female mice were used for experimental analysis. Experiments involving animals were conducted in accordance with institutional guidelines and following the protocols approved by the Institutional Animal Care and Use Committee of Sichuan Provincial People's Hospital (approval number: NSFC2014-009).

## Preparation of flat-mount retinae and retina sections

Enucleated eyes were fixed for 15 min in 4% paraformaldehyde (PFA) at room temperature, followed by immersion in phosphate buffered saline (PBS) for the same duration. The retinae were then dissected and partially cut into four quadrants as previously described (*Liu et al., 2022*). The dissected retinal cups were post-fixed overnight and then stained as described below. As for retina sections, after fixation in 4% PFA for 2 hr, the enucleated eyes were dehydrated until they settled to the bottom of the tube with 30% sucrose and then embedded in Tissue Freezing Medium (Servicebio). Specimens were sectioned, cleaned, and stained as previously described (*Yang et al., 2019*).

## Hyaloid vessel preparation

Neonatal eyes were fixed in 4% PFA for 2 hr before the cornea, lens, and iris were removed. The eye cup was immersed in 5% gelatin at 4°C overnight. The solidified gelatin was extricated and heat melted on a glass slide, washed with warm water, air-dried, and imaged with DAPI staining.

## Immunohistochemistry

All immunostainings of retinae were performed using littermates with similar body size and treated under the same conditions. The flat-mount retina and retina sections were permeabilized and blocked in PBS containing 5% fetal bovine serum and 0.2% Triton X-100 at room temperature for 2 hr. Next, they were rinsed in PBS and incubated in directly conjugated IB4 Alexa 488 or Alexa 594 (1:200; Thermo Fisher, USA) and primary antibodies in a blocking buffer at 4°C overnight. After three washes, the retinae were processed for multiple labeling or flat mounted onto microscope glass slides with Fluoromount (Sigma-Aldrich, USA).

For in vivo analysis of cell proliferation, each pup was injected intraperitoneally with 200 µg EdU 3 hr before the mice were euthanasia. Retinae were dissected and permeabilized as previously described. EDU-positive labeling was stained and detected by means of a Click-iT EDU Alexa Fluor-647 Imaging Kit (C10640, Thermo Fisher Scientific, USA) according to the manufacturer's instruction. For para-cellular BRB integrity analysis, each pup was injected intraperitoneally with 2% fluorescent tracer, 5-(and-6)-tetramethylrhodamine biocytin, biocytin-TMR (CAT#T12921, Thermo Fisher Scientific, USA) for 24 hr prior to sacrifice, followed by flat-mount preparation, blocking, and staining of Isolectin B4.

Images of stained flat-mounted retinae, retina sections, and HRECs were acquired using Zeiss LSM 800 confocal microscope (Thornwood, NY, USA) and processed with Zeiss processing software, Angio Tool, and Adobe Photoshop. The detailed information about antibodies used for immunofluorescence is listed in *Supplementary file 4*.

## Cell lines and primary cell culture

HRECs were purchased from Cell System and cultured in EBM-2 media (CC-3156, Lonza, Switzerland) containing 5% fetal bovine serum, 0.4% hydrocortisone, 4% Hfgf-b, 0.1% vascular endothelial growth factor (VEGF), 0.1% R3-IGF, 0.1% ascorbic acid, 0.1% human epidermal growth factor (hEGF) 0.1% GA-1000, and 0.1% heparin at 37°C in a 5% $CO_2$ incubator. HRECs at passages 3–7 were used for experiments.

HEK-293T and HEK-293STF were obtained from the American Type Culture Collection (ATCC) and cultured in Dulbecco's Modified Eagle Medium (DMEM) (SH30023.01, Hycolon, USA) with 10% fetal bovine serum and cells were maintained at 37°C in a 5% $CO_2$ incubator. Both cell lines were authenticated by STR profiling, and no mycoplasma contamination was found.

## Gene knockdown and overexpression strategies

HRECs were seeded on 6 cm dishes at the day before transfected with lentivirus carrying shRNA for human *CAPSL* (5'-AAGACCTTCGTGAAGTATA-3', Genechem, Shanghai, China) or negative control shRNA (5'-TTCTCCGAACGTGTCACGT-3') according to the protocol of the manufacture. Cells were incubated for at least 72 hr before used for further experiments.

For overexpression of exogenous protein in HEK-293T cells, cells were transfected with Lipofectamine-3000 (Invitrogen, USA) according to the manufacturer's instructions.

## Matrigel EC tube formation assay

HRECs transfected with the corresponding shRNA after 72 hr were counted and planted onto the matrigel as previously described (*Zhang et al., 2019*). After 6 hr of incubation at 37°C in a 5% $CO_2$ incubator, the images were captured under an anatomical lens (Carl Zeiss, Germany).

## Immunofluorescence of HREC

After cell counting, the same number of HRECs transfected with shCtrl and shCAPSL was seeded on slides in 24-well plates. Cells were fixed with 4% PFA at room temperature for 15 min. After being rinsed in PBS three times, cells were blocked in the blocking buffer at room temperature for 2 hr. The cells were labeled with primary antibodies overnight at 4°C. Following three washes in PBS, Alexa-Fluor-594- or Alexa-Fluor-647-conjugated antibodies were administered along with DAPI, and cells were incubated at room temperature for 1 hr. After three washes in PBS, stained cells were mounted on microscope glass slides with Fluoromount (Sigma-Aldrich, USA).

For in vitro analysis of cell proliferation, 20 μM EdU (Thermo Fisher Scientific, USA) was added into 96-well plated 3 hr before cells were harvested. EdU-positive cells were stained and detected with the Click-iT EDU Alexa Fluor-647 Imaging Kit (C10640, Thermo Fisher Scientific, USA) according to the manufacturer's instruction.

## Wound healing assay

Seventy-two hours after shRNA transfection, HRECs were seeded in 6-well plates with complete medium for 24 hr until grown to confluence. Plates were then coated with 10 μg/ml fibronectin for half an hour in 37°C before cell culture. 200 μl pipette tips were used to generate wounds, and each well was washed twice with Utrasalin A (Lonza, Switzerland) to remove detached cells. Cells were then starved in the EBM-2 medium at 37°C in a 5% $CO_2$ incubator. Images of the wounds were captured with an optical microscope at 0, 12, and 16 hr after the wounds were made, respectively. The wound closure areas were measured using ImageJ software.

For analyses of Golgi apparatus polarization, cells were maintained on 24-well glass slides and fixed with 4% PFA for 15 min at 9 hr after cell migration was initiated.

## RNA extraction and RT-qPCR

Total RNA was extracted from cultured HRECs or fresh lungs from mouse using TRIzol reagent (Life Technologies, USA) according to the manufacturer's instructions. RNA concentration and quality were measured with NanoDrop 2000 spectrophotometer (Thermo Fisher Scientific, USA), and 1 μg total RNA was reverse transcribed using EasyScript One-Step RT-PCR SuperMix (TransGen Biotech, Beijing, China) following the standard protocol. RT-qPCR was performed using a 7500 Real-Time PCR System (Applied Biosystems, USA) with TransStart Tip Green qPCR Supermix (TransGen Biotech). At least three experiments were performed, and PCR reactions were performed in triplicate. Glyceraldehyde-3-Phosphate Dehydrogenase (GAPDH) was used as a reference gene. The primer sets are listed in *Supplementary file 5*. All experiments were replicated three times and representative results were shown.

## Protein extraction and western blotting

HRECs or lung tissue from mice were lysed in standard radioimmunoprecipitation assay buffer (PIPA, 50 mM Tris–HCl, 150 mM NaCl, 1% Triton X-100, 0.5% sodium deoxycholate, 0.1% sodium dodecyl sulfate, pH 7.4) supplemented with Protease and Phosphatase Inhibitor Cocktail (Roche, USA). After sonication, centrifugation, and protein quantification, the supernatant was diluted in 2× sodium dodecyl sulfate (SDS) loading buffer. Equal amounts of protein (20 μg) were loaded and resolved

on 10–15% SDS–polyacrylamide, and western blotting was performed. After gels were transfected onto a polyvinylidene difluoride membrane (GE Healthcare, USA), the membranes were incubated in a blocking buffer containing 8% skimmed milk (9999, Cell Signaling Technology) in tris buffered saline with Tween 20 (TBST) for 1 hr at room temperature on a rocking platform, followed by overnight incubation at 4°C with the primary antibodies diluted in blocking buffer. Membranes were then washed in TBST three times and incubated in the blocking buffer containing secondary Horseradish Peroxidase-conjugated antibodies for 1 hr at room temperature. Protein blots were visualized with e-BLOT Touch Imager (e-BLOT, China) and blots were relatively quantified with ImageJ. GAPDH was used as an internal reference. Detailed information about antibodies used for western blot is listed in *Supplementary file 4*. All experiments were replicated three times and representative blot images were shown.

## Luciferase assays

Dual-luciferase assays were performed as previously described (*Zhu et al., 2021*). Briefly, 293STF cells were plated on 24-well plates at a confluency of ~30%. The following day, the cells were transfected with corresponding plasmids with lip3000. Forty-eight hours post-transfection, cells were harvested, and the lysates were used to measure Firefly and *Renilla* luciferase activity according to the manufacturer's instructions (TransGen, Bejing, China).

## Quantification of retinal parameters

All quantifications were done using Zeiss processing software on high-resolution confocal images. Vascular progression was measured in a straight line from the optic nerve to the angiogenic front of the retinal plexus for each retina quadrant ($n = 6$, each group). Vessel density and branch points were calculated with Angio Tool software from flat-mounted retinae ($n = 6$, each group). Endothelial tip cell numbers were measured by counting endothelial sprouts at the angiogenic front of the entire vascular plexus of the same length ($n = 6$, each group). Endothelial tip cell filopodia density was calculated with high-resolution images (×60 objective) of six randomly selected angiogenic front fields from six retinae per genotype ($n = 6$, each group). EC proliferation was calculated by measuring the number of both EDU- and ERG-positive cells within retinae of the same size and same position at the edge of the vascular plexus ($n = 6$, each group). Retinal pericyte coverage was quantified by comparing the $NG2^+$ or $Desmin^+$ pericytes with $IB4^+$ vessels ($n = 6$, each group). The nuclear ellipticity of the cells at the scratch edge was calculated by the distance of the nuclear long axis (Height) divided by the maximum vertical distance to the nuclear long axis (Width).

## Transcriptomic profiling

Total RNA was extracted from shCtrl-ECs, sh*CAPSL*-ECs, and LentiOE-ECs and quantified using NanoDrop2000 Spectrophotometer (Thermo Fisher Scientific, USA). RNA integrity was determined by Bioanalyzer RNA 6000 Nano assay kit (Agilent, China). RNA library construction was performed with Illumina TruseqTM RNA sample prep Kit and RNA-seq was performed with Truseq SBS Kit (300 cycles) in BIOZERON (BIOZERON, China). RNA-seq reads were performed using the TopHat software tool to acquire the alignment file, which was used to quantify mRNA expression and determine the differentially expressed genes. All gene expression values were changed to log2 values for further analysis, and a p-value of less than or equal to 0.05 was considered to indicate significance. The GSEA was performed with the omicshare platform (https://www.omicshare.com). The raw data have been uploaded to the Genome Sequence Archive (https://ngdc.cncb.ac.cn/gsa/browse/HRA010305), and the assigned accession number was HRA010305.

## Proteomic profiling

Total protein was extracted from shCtrl-ECs, sh*CAPSL*-ECs, and LentiOE-ECs and quantified using a BCA protein quantification kit (Beyotime, China). Samples were sent to PTM Biolab (PTM Biolab, China) to perform proteomic profiling using 4D-Labelfree. All protein expression values were changed to log2 values for further analysis, with a corrected p-value <0.05. The GSEA was performed with the omicshare platform (https://www.omicshare.com). The raw data have been uploaded to the Genome Sequence Archive (https://www.ebi.ac.uk/pride/archive), and the assigned accession number was PXD051696.

## Image acquisition and statistical analysis

Immunofluorescence images were obtained using a laser scanning microscope (LSM 900, Zeiss). Statistical analyses were performed with GraphPad Prism 8.0 software. Data were analyzed using the Student's $t$ test for comparison between two groups or one-way analysis of variance followed by Dunnett multiple comparison test for comparison of multiple groups, and plotted as mean ± SD. At least three independent experiments were performed. ImageJ was used to compare results for the tube formation assay experiments and bands of immunoblot. Results were considered significant when $*p<0.05$, $**p<0.01$, $***p<0.001$, or $****p<0.0001$. 'ns' stands for 'no significance'.

## Study approval

This study was approved by the institutional review board of Sichuan Provincial People's Hospital (approval number: NSFC2014-009) and Xinhua Hospital Affiliated to Shanghai Jiao Tong University School of Medicine, and informed consent was obtained from all participants or legal guardians for minors. All animal studies were performed according to established ethical guidelines approved by the animal care committee of Sichuan Provincial People's Hospital (approval number: NSFC2014-009).

## Additional information

### Funding

| Funder | Grant reference number | Author |
|---|---|---|
| National Natural Science Foundation of China | 82371083 | Xianjun Zhu |
| National Natural Science Foundation of China | 82121003 | Zhenglin Yang |
| Sichuan Province Science and Technology Support Program | 2023ZYD0172 | Xianjun Zhu |
| Sichuan Provincial People's Hospital Postdoctoral fund | 2022BH019 | Wenjing Liu |
| Sichuan Provincial People's Hospital Huanhua Talent program | | Xianjun Zhu |
| National Natural Science Foundation of China | 82101153 | Mu Yang |
| CAMS Innovation Fund for Medical Sciences | | Zhenglin Yang |
| Sichuan Intellectual Property Office (China) | | Xianjun Zhu |

The funders had no role in study design, data collection, and interpretation, or the decision to submit the work for publication.

### Author contributions

Wenjing Liu, Conceptualization, Data curation, Investigation, Methodology, Writing – original draft, Writing - review and editing; Shujin Li, Data curation, Validation, Investigation; Mu Yang, Data curation, Validation, Visualization, Methodology, Writing – original draft; Jie Ma, Validation, Investigation; Lu Liu, Investigation, Methodology; Ping Fei, Qianchun Xiang, Lulin Huang, Validation, Methodology; Peiquan Zhao, Conceptualization, Data curation, Supervision, Validation, Investigation, Project administration; Zhenglin Yang, Conceptualization, Resources, Data curation, Supervision, Funding acquisition, Validation, Writing – original draft, Project administration; Xianjun Zhu, Conceptualization, Resources, Data curation, Formal analysis, Supervision, Funding acquisition, Validation, Investigation, Writing – original draft, Project administration, Writing - review and editing

## Author ORCIDs
Wenjing Liu http://orcid.org/0000-0001-8037-6275
Mu Yang https://orcid.org/0000-0002-6001-6827
Jie Ma https://orcid.org/0000-0002-3746-4381
Lulin Huang https://orcid.org/0000-0002-1204-5957
Xianjun Zhu https://orcid.org/0000-0002-2531-7552

## Ethics

The study was approved by the Institutional Research Committees of Sichuan Provincial People's Hospital (approval number: NSFC2014-009). Informed consent was obtained from all participants and from guardians of minors involved in genetic testing. The patients were clinically diagnosed with exudative vitreoretinopathy. Family ID numbers 3036 and 3104 are not known to anyone outside the research group.

All experiments involving animals were conducted in accordance with institutional guidelines and following the protocols approved by the Institutional Animal Care and Use Committee of Sichuan Provincial People's Hospital (approval number: NSFC2014-009).

Reviewer #1 (Public Review): https://doi.org/10.7554/eLife.96907.3.sa1
Reviewer #2 (Public Review): https://doi.org/10.7554/eLife.96907.3.sa2
Reviewer #3 (Public Review): https://doi.org/10.7554/eLife.96907.3.sa3
Author response https://doi.org/10.7554/eLife.96907.3.sa4

---

# Additional files

## Supplementary files
- Supplementary file 1. The results of pathological prediction of CAPSL variants.
- Supplementary file 2. The differentially regulated genes in shCtrl-ECs and shCAPSL-ECs.
- Supplementary file 3. Down-regulated genes shared in *CAPSL*-depleted HRECs and *MYC*-depleted HUVECs.
- Supplementary file 4. List of the antibodies used in this study.
- Supplementary file 5. List of the PCR primers used in this study.
- MDAR checklist

## Data availability

All data produced or analyzed in the present study are included in the manuscript and supplementary files. RNAseq data and Proteomic profiling data of shCtrl-ECs, shCAPSL-ECs, and LentiOE-ECs have been uploaded to the Genome Sequence Archive (https://ngdc.cncb.ac.cn/gsa-human/browse/HRA007150) and EMBL-EBI Archive (https://www.ebi.ac.uk/pride/archive/PXD051696).

The following datasets were generated:

| Author(s) | Year | Dataset title | Dataset URL | Database and Identifier |
|---|---|---|---|---|
| Liu W, Li S, yang M, Ma J, Liu L, Fei P, Xiang Q, Huang L, Zhenglin Y, Zhu X | 2024 | RNA-seq raw data for CAPSL HRECs | https://ngdc.cncb.ac.cn/gsa-human/browse/HRA007150 | Genome Sequence Archive, HRA007150 |
| Liu W, Li S, Yang M, Ma J, Liu L, Fei P, Xiang Q, Huang L, Zhao P, Yang Z, Zhu X | 2024 | Proteomic raw data for ctrl HRECs, CAPSL knockdown HRECs, and CAPSL overexpression HRECs | https://www.ebi.ac.uk/pride/archive/projects/PXD051696 | PRIDE, PXD051696 |

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
