## [Editor Report · eLife assessment]

This study explores the role of calcyphosine-like (CAPSL) in Familial Exudative Vitreoretinopathy (FEVR) via the MYC pathway, offering **valuable** insights into disease mechanisms that are supported by a solid, multi-pronged approach. The manuscript, which presents the phenotype of an interesting new mouse model, provides **convincing** evidence that CAPSL variants cause disease.

---

## [Referee Report · Reviewer #1 (Public Review)]

Summary:

The author presents the discovery and characterization of CAPSL as a potential gene linked to Familial Exudative Vitreoretinopathy (FEVR), identifying one nonsense and one missense mutation within CAPSL in two distinct patient families afflicted by FEVR. Cell transfection assays suggest that the missense mutation adversely affects protein levels when overexpressed in cell cultures. Furthermore, conditionally knocking out CAPSL in vascular endothelial cells leads to compromised vascular development. The suppression of CAPSL in human retinal microvascular endothelial cells results in hindered tube formation, a decrease in cell proliferation, and disrupted cell polarity. Additionally, transcriptomic and proteomic profiling of these cells indicates alterations in the MYC pathway.

Strengths:

The study is nicely designed with a combination of in vivo and in vitro approaches, and the experimental results are good quality.

Weaknesses:

My reservations lie with the main assertion that CAPSL is associated with FEVR, as the genetic evidence from human studies appears relatively weak. Further careful examination of human genetics evidence in both patient cohorts and the general population will help to clarify. In light of human genetics, more caution needs to be exercised when interpreting results from mice and cell model and how is it related to the human patient phenotype. Future replication by finding more FEVR patients with a mutation in CAPSL will strengthen the findings.

---

## [Referee Report · Reviewer #2 (Public Review)]

Summary:

This work identifies two variants in CAPSL in two generation familial exudative vitreoretinopathy (FEVR) pedigrees, and using a knockout mouse model, they link CAPSL to retinal vascular development and endothelial proliferation through the MYC pathway. Together, these findings suggest that the identified variants may be causative and that CAPSL is a new FEVR-associated gene.

Strengths:

The authors data provides compelling evidence that loss of the poorly understood protein CAPSL can lead to reduced endothelial proliferation in mouse retina and suppression of MYC signaling, consistent with the disease seen in FEVR patients. The paper is clearly written, and the data generally support the author's hypotheses.

Weaknesses:

(1) Both pedigrees described suggest autosomal dominant inheritance in humans, but no phenotype was observed in Capsl heterozygous mice. Additional studies would be needed to determine the cause of this disparity.

(2) Additional discussion of the hypothesized functional mechanism of the p.L83F variant would have improved the manuscript. While the human genetic data is compelling, it remains unclear how this variant may effect CAPSL function. In vitro, p.L83F protein appears to be normally localized within the cell and it is unclear why less mutant protein was detected in transfected cells. Was the modified protein targeted for degradation?

(3) Authors did not describe how the new crispr-generated Capsl-loxp mouse model was screened for potential off-target gene editing, raising the possibility that unrelated confounding mutations may have been introduced.

---

## [Referee Report · Reviewer #3 (Public Review)]

Summary:

This manuscript by Liu et al. presents a case that CAPSL mutations are a cause of familial exudative vitreoretinopathy (FEVR). Attention was initially focused on the CAPSL gene from whole exome sequence analysis of two small families. The follow-up analyses included studies in which Capsl was manipulated in endothelial cells of mice and multiple iterations of molecular and cellular analyses. Together, the data show that CAPSL influences endothelial cell proliferation and migration. Molecularly, transcriptomic and proteomic analyses suggest that CAPSL influences many genes/proteins that are also downstream targets of MYC and may be important to the mechanisms.

Strengths:

This multi-pronged approach found a previously unknown function for CAPSL in endothelial cells and pointed at MYC pathways as high-quality candidates in the mechanism. Through the review process, some statements and interpretations were initially challenged. However, the issues were addressed with new experimentation and modifications to the text - leaving a strengthened presentation that makes a compelling case.

Weaknesses:

Two issues shape the overall impact for me. First, it remains unclear how common CAPSL variants may be in the human population. From the current study, it is possible that they are rare - perhaps limiting an immediate clinical impact. However, sharing the data may help identify additional variants in FEVR or other vascular diseases. The findings also make advances in basic biology which could ultimately contribute to therapies of broad relevance. Thus, this weakness is considered modest. Second, the links to the MYC axis are largely based on association, which will require additional experimentation to help understand.

One interesting technical point raised in the study, which might be missed without care by the readership, is that the variants appear to act dominantly in human families, but only act recessively in the mouse model. The authors cite other work from the field in which this same mismatch occurs, likely pointing to limits in how closely a mouse model might be expected to recapitulate a human disease. This technical point is likely relevant to ongoing studies of FEVR and many other multigenic diseases as well.

---

## [Author Response]

The following is the authors’ response to the original reviews.

**Reviewer #1 (Public Review):**
Summary:The author presents the discovery and characterization of CAPSL as a potential gene linked to Familial Exudative Vitreoretinopathy (FEVR), identifying one nonsense and one missense mutation within CAPSL in two distinct patient families afflicted by FEVR. Cell transfection assays suggest that the missense mutation adversely affects protein levels when overexpressed in cell cultures. Furthermore, conditionally knocking out CAPSL in vascular endothelial cells leads to compromised vascular development. The suppression of CAPSL in human retinal microvascular endothelial cells results in hindered tube formation, a decrease in cell proliferation, and disrupted cell polarity. Additionally, transcriptomic and proteomic profiling of these cells indicates alterations in the MYC pathway.Strengths:The study is nicely designed with a combination of in vivo and in vitro approaches, and the experimental results are good quality.

We thank the reviewer for the conclusion and positive comments.

Weaknesses:My reservations lie with the main assertion that CAPSL is associated with FEVR, as the genetic evidence from human studies appears relatively weak. Further careful examination of human genetics evidence in both patient cohorts and the general population will help to clarify. In light of human genetics, more caution needs to be exercised when interpreting results from mice and cell models and how is it related to the human patient phenotype.

We thank the reviewer for careful reading and constructive suggestion. we added several experiments to address the concern of reviewer are as follows:

(1) The pLI score of LOF allele of CAPSL is based of general population, among which Europeans account for ~77% and East Asians make up less than 3%. Since the FEVR families in this article all come from China, the pLI score may not be accurate. Of course, we will continue to collect FEVR pedigrees.

(2) We evaluated the phenotype of *Capsl* heterozygous mice at P5, and the results showed no overt difference in vascular progression, vessel density and branchpoints with littermate wildtype controls (Fig.S4). The lack of pronounced phenotype in FEVR heterozygous mice may be due to different sensitivity between human and mice. A similar example is LRP5 mutations associated with FEVR. Heterozygous mutations in LRP5 were reported in FEVR patients in multiple populations (PMID: 16929062, 33302760, 27486893, 35918671, 36411543). However, heterozygous Lrp5 knockout mice exhibited no visible angiogenic phenotype (PMID: 18263894). Corresponding description was added in the manuscript at page 6.

(3) We further assessed the angiogenic phenotype when angiogenesis almost complete at P21, and the resulted revealed no difference observed between Ctrl and CapsliECKO/iECKO mice (Fig.S5). And corresponding description was added in the manuscript at page 7.

(4) We evaluated the expression of MYC downstream genes in vivo using lung tissue form P35 Ctrl and _Capsl_iECKO/iECKO mice (Fig.S8). Consistent with the results from in vitro HRECs, _Capsl_iECKO/iECKO mice showed downregulated expression of MYC targets. And corresponding description was added in the manuscript at page 11.

**Reviewer #2 (Public Review):**
Summary:This work identifies two variants in CAPSL in two-generation familial exudative vitreoretinopathy (FEVR) pedigrees, and using a knockout mouse model, they link CAPSL to retinal vascular development and endothelial proliferation. Together, these findings suggest that the identified variants may be causative and that CAPSL is a new FEVR-associated gene.Strengths:The authors' data provides compelling evidence that loss of the poorly understood protein CAPSL can lead to reduced endothelial proliferation in mouse retina and suppression of MYC signaling in vitro, consistent with the disease seen in FEVR patients. The study is important, providing new potential targets and mechanisms for this poorly understood disease. The paper is clearly written, and the data generally support the author's hypotheses.

We thank the reviewer for the conclusion and positive comments.

Weaknesses:(1) Both pedigrees described appear to suggest that heterozygosity is sufficient to cause disease, but authors have not explored the phenotype of Capsl heterozygous mice. Do these animals have reduced angiogenesis similar to KOs? Furthermore, while the p.R30X variant protein does not appear to be expressed in vitro, a substantial amount of p.L83F was detectable by western blot and appeared to be at the normal molecular weight. Given that the full knockout mouse phenotype is comparatively mild, it is unclear whether this modest reduction in protein expression would be sufficient to cause FEVR - especially as the affected individuals still have one healthy copy of the gene. Additional studies are needed to determine if these variants alter protein trafficking or localization in addition to expression, and if they can act in a dominant negative fashion.

We thank the reviewer for the suggestion. We evaluated the phenotype of Capsl heterozygous mice at P5 (Fig.S4), and the results showed no overt difference in angiogenesis compared with littermate control mice.

We transfected CAPSL wild-type plasmid, p.R30X mutant plasmid and p.L83F mutant plasmid into 293T cells to assess the intracellular localization change of CAPSL mutant proteins (Fig.S1). The result showed that the point mutation did not affect the localization of the mutated protein, and corresponding description was added in the manuscript at page 5.

(2) The manuscript nicely shows that loss of CAPSL leads to suppressed MYC signaling in vitro. However, given that endothelial MYC is regulated by numerous pathways and proteins, including FOXO1, VEGFR2, ERK, and Notch, and reduced MYC signaling is generally associated with reduced endothelial proliferation, this finding provides little insight into the mechanism of CAPSL in regulating endothelial proliferation. It would be helpful to explore the status of these other pathways in knockdown cells but as the authors provide only GSEA results and not the underlying data behind their RNA seq results, it is difficult for the reader to understand the full phenotype. Volcano plots or similar representations of the underlying expression data in Figures 6 and 7 as well as supplemental datasets showing the differentially regulated genes should be included. In addition, while the paper beautifully characterizes the delayed retinal angiogenesis phenotype in CAPSL knockout mice, the authors do not return to that model to confirm their in vitro findings.

We thank the reviewer for the suggestion. Although endothelial MYC can be regulated by FOXO1, VEGFR2, ERK, and Notch signaling pathway, these pathways are not enriched in the RNA seq data of CAPSL-depleted HRECs. This suggests that the down regulated MYC targets may not be influenced by the signaling pathway mentioned above. RNA-seq raw data have been uploaded to the Genome Sequence Archive (https://ngdc.cncb.ac.cn/gsa/browse/HRA010305) and proteomic profiling raw data have been uploaded to the Genome Sequence Archive (https://www.ebi.ac.uk/pride/archive), and the assigned accession number was PXD051696. Corresponding description was added in the manuscript at page 20-21. The datasets represent the differentially regulated genes in Figure 6 and 7 were listed at Dataset S1 and S2.

(3) In Figure S2D, the result of this vascular leak experiment is unconvincing as no dye can be seen in the vessels. What are the kinetics for biocytin tracers to enter the bloodstream after IP injection? Why did the authors choose the IP instead of the IV route for this experiment? Differences in the uptake of the eye after IP injection could confound the results, especially in the context of a model with vascular dysfunction as here.

We thank the reviewer for suggestion. In Figure S2D (now Fig.S6D), we used a non-representative image to show vascular leakage. We replaced the images with more representative ones. We are sorry that we are not clear about the kinetics for biocytin tracers to enter the bloodstream after IP injection. Since the experiment was carried out on mice at P5, it is not feasible to do IV injection in P5 neonatal mice. We followed the methods described in the previous study involving mice of same age (PMID:35361685).

(4) In Figure 5, it is unclear how filipodia and tip cells were identified and selected for quantification. The panels do not include nuclear or tip cell-specific markers that would allow quantification of individual tip cells, and in Figure 5C it appears that some filipodia are not highlighted in the mutant panel.

We thank the reviewer for the comments. In Figure 5, we used HRECs to examine the cell proliferation, migration and polarity in vitro, and therefore there is no distinction between tip cells and stalk cells. The quantification of filopodia/lamellipodia was performed as previous studies (PMID: 30783090, PMID: 28805663). In briefly, wound scratch was performed on confluent layers of transfected HRECs, and 9 hours after initiating cell migration by scratch, cells were fixed and stained with phalloidin. Cells at the edge of wound were considered as leader cells and quantified for number of filopodia/lamellipodia.

**Reviewer #3 (Public Review):**
Summary:This manuscript by Liu et al. presents a case that CAPSL mutations are a cause of familial exudative vitreoretinopathy (FEVR). Attention was initially focused on the CAPSL gene from whole exome sequence analysis of two small families. The follow-up analyses included studies in which CAPSL was manipulated in endothelial cells of mice and multiple iterations of molecular and cellular analyses. Together, the data show that CAPSL influences endothelial cell proliferation and migration. Molecularly, transcriptomic and proteomic analyses suggest that CAPSL influences many genes/proteins that are also downstream targets of MYC and may be important to the mechanisms.Strengths:This multi-pronged approach found a previously unknown function for CAPSLs in endothelial cells and pointed at MYC pathways as high-quality candidates in the mechanism.Weaknesses:Two issues shape the overall impact for me. First, the unreported population frequency of the variants in the manuscript makes it unclear if CAPSL should be considered an interesting candidate possibly contributing to FEVR, or possibly a cause. Second, it is unclear if the identified variants act dominantly, as indicated in the pedigrees. The studies in mice utilized homozygotes for an endothelial cell-specific knockout, leaving uncertainty about what phenotypes might be observed if mice heterozygous for a ubiquitous knockout had instead been studied.In my opinion, the following scientific issues are specific weaknesses that should be addressed:(1) Please state in the manuscript the number of FEVR families that were studied by WES. Please also describe if the families had been selected for the absence of known mutations, and/or what percentage lack known pathogenic variants.

We thank the reviewer for thoughtful comments. 120 FEVR families were studied by WES and we added corresponding description in the manuscript at page 4.

(2) A better clinical description of family 3104 would enhance the manuscript, especially the father. It is unclear what "manifested with FEVR symptoms, according to the medical records" means. Was the father diagnosed with FEVR? If the father has some iteration of a mild case, please describe it in more detail. If the lack of clinical images in the figure is indicative of a lack of medical documentation, please note this in the manuscript.

We thank the reviewer for thoughtful comments. The father of family 3104 has also been identified as a carrier of this heterozygous variant, manifested with FEVR symptoms, according to the medical records. Nevertheless, clinical examination images are presently unavailable. We added corresponding description in the manuscript at page 5.

(3) The TGA stop codon can in some instances also influence splicing (PMID: 38012313). Please add a bioinformatic assessment of splicing prediction to the assays and report its output in the manuscript.

We thank the reviewer for thoughtful comments. We predicted the splicing of c.88C>T variant of CAPSL using MaxEntScan (http://hollywood.mit.edu/burgelab/maxent/Xmaxentscan_scoreseq.html) and SpliceTool (https://rddc.tsinghua-gd.org/ai) (Fig.S2). MaxEntScan and SpliceTool were used to predict the impact of TGA stop codon of c.88C>T variant on the formation of a cryptic donor splice site.

(4) More details regarding utilizing a "loxp-flanked allele of CAPSL" are needed. Is this an existing allele, if so, what is the allele and citation? If new (as suggested by S1), the newly generated CAPSL mutant mouse strain needs to be entered into the MGI database and assigned an official allele name - which should then be utilized in the manuscript and who generated the strain (presumably a core or company?) must be described.

We added detailed description of Capsl flxoed allele to Method section on page 14-15: “Capslloxp/+ model was generated using the CRISPR/Cas9 nickase technique by Viewsolid Biotechology (Beijing, China) in C57BL/6J background and named Capslem1zxj. The genomic RNA (gRNA) sequence was as follows: Capsl-L gRNA: 5’-CTATCCCAA TTGTGCTCCTGG-3’; Capsl-R gRNA: 5’-TGGGACTCATGGTTCTAGAGG-3’. ”

(5) The statement in the methods "All mice used in the study were on a C57BL/6J genetic background," should be better defined. Was the new allele generated on a pure C57BL/6J genetic background, or bred to be some level of congenic? If congenic, to what generation? If unknown, please either test and report the homogeneity of the background, or consult with nomenclature experts (such as available through MGI) to adopt the appropriate F?+NX type designation. This also pertains to the Pdgfb-iCreER mice, which reference 43 describes as having been generated in an F2 population of C57BL/6 X CBA and did not designate the sub-strain of C57BL/6 mice. It is important because one of the explanations for missing heritability in FEVR may be a high level of dependence on genetic background. From the information in the current description, it is also not inherently obvious that the mice studied did not harbor confounding mutations such as rd1 or rd8.

We thank the reviewer for suggestion. We added the following description to “Mouse model and genotyping” method section on page 14. “Capslloxp/+ model was generated using the CRISPR/Cas9 nickase technique by Viewsolid Biotechology (Beijing, China) in C57BL/6J background and named Capslem1zxj. The genomic RNA (gRNA) sequence was as follows: Capsl-L gRNA: 5’-CTATCCCAA TTGTGCTCCTGG-3’; Capsl-R gRNA: 5’-TGGGACTCATGGTTCTAGAGG-3’. Pdgfb-iCreER[43] transgenic mice on a mixed background of C57BL/6 and CBA was obtainted from Dr. Marcus Fruttiger and backcrossed to background for 6 generations. Capslloxp/+ mice were bred with Pdgfb-iCreER[43] transgenic mice to generate Capslloxp/loxp, Pdgfb-iCreER mice.” Sanger sequencing was performed on experimental mice to identify whether they harbor confounding mutations such as Pde6b or Crb1. The results showed the mice did not harbor confounding mutations (Fig.S9) and corresponding description was added in the manuscript at page 15.

(6) In my opinion, more experimental detail is needed regarding Figures 2 and 3. How many fields, of how many retinas and mice were analyzed in Figure 2? How many mice were assessed in Figure 3?

We thank the reviewer for thoughtful comments. We have already presented the detailed information in the manuscript, please refer to the “Methods-Quantification of retinal parameters” section for experimental details.

(7) I suggest adding into the methods whether P-values were corrected for multiple tests.

We thank the reviewer for suggestion. Actually, the statistical analysis was performed using unpaired Student’s t-test for comparison between two groups or one-way ANOVA followed by Dunnett multiple comparison test for comparison of multiple groups. The above description was added to “Methods-Image acquisition and statistical analysis” section to make it clear.

**Recommendations for the authors:**

**Reviewing Editor (Recommendations For The Authors):**
In summary, the following concerns should addressing reviewers' concerns as outlined below could bolster the evidence from "solid" to "convincing" and further strengthen the study's impact.(1) Analysis of the phenotype in CAPSLheterozygous mice, as highlighted by all 3 reviews.We thank the editor for thoughtful comments. The phenotype analysis of *Capsl* heterozygous mice was added to Fig.S4, with the corresponding description provided at page 6.(2) Analysis of Capsl KO mice to determine if the pathways identified in vitro are modified (as suggested by reviewers 1 & 2).

We thank the editor for suggestion. In Fig.S7, RT-qPCR was performed on lung tissues from *Capsl* Ctrl and KO mice to validate the expression of MYC targets in vivo. And the result indicated that the downstream targets of MYC signaling were also downregulated in vivo, consistent with the *in vitro* findings.

(3) Additional description of the genetic pedigrees and variants to address the points raised by reviewer #3.

We thank the editor for suggestion. The father of family 3104 has also been identified as a carrier of this heterozygous variant, manifested with FEVR symptoms, according to the medical records. Nevertheless, clinical examination data are presently unavailable. We added corresponding description in the manuscript page 5.

(4) Validation of the identified protein variants, especially L83F which appears to be expressed at a near normal level. Are these proteins mislocalized, do the variants to interfere with sites of known or predicted protein-protein interactions, could they act in a dominant-negative fashion by aggregation with co-expressed WT protein etc. Given the comparatively weak genetic data, additional validation is required to establish plausibility of CAPSL as a FEVR gene.

We thank the editor for suggestion. As substantial amount of p.L83F was detectable at normal molecular weight, we further investigated whether this variant affects protein localization. Fig.S1, immunocytochemistry results indicated that this variant does not affect the subcellular localization of the protein.

(5) Improved description of experimental details and statistical analyses as outlined by reviewer #3.

We thank the editor for suggestion. The more detailed information about *Capsl* mice was added in the manuscript at page 14-15. The experimental details regarding Figure 2 and Figure 3 have already presented in the “Methods-Quantification of retina parameters” section in the manuscript at page 19-20. And the statistical analysis was performed using unpaired Student’s t-test for comparison between two groups or one-way ANOVA followed by Dunnett multiple comparison test for comparison of multiple groups. The above description was added to “Methods-Image acquisition and statistical analysis” section at page 21 to make it clear.

**Reviewer #1 (Recommendations For The Authors):**
My reservations lie with the main assertion that CAPSL is associated with FEVR, as the genetic evidence from human studies appears relatively weak. My concerns are as follows:(1) The molecular characterization of the identified mutations suggests a loss of function (LOF). Notably, in one family, both the father and son exhibit the FEVR phenotype and share the LOF mutation, suggesting a dominant mode of inheritance. However, the prevalence of the LOF allele of CAPSL in the general population is high, and its pLI score is 0, according to the GNOMAD database. This raises doubts about the LOF variant of CAPSL being causative for FEVR.

We thank the reviewer for recommendation. The pLI score of LOF allele of CAPSL is based of general population, among which Europeans account for ~77% and East Asians make up less than 3%. Since the FEVR families in this article all come from China, the pLI score may not be accurate. Of course, we will continue to collect FEVR pedigrees and screen for CAPSL mutations.

(2) In the conditional knockout study, a delay in vascular development is observed in the retina up to P14. What the phenotype looks like in adult mice and whether it replicates the human FEVR phenotype?

We thank the reviewer for recommendation. We further assessed the phenotype when angiogenesis almost complete at P21, the resulted showed no difference in Ctrl and CapsliECKO/iECKO mice (Fig.S5). And corresponding description was added in the manuscript at page 7.

(3) The conditional knockout mice lack both alleles of CAPSL. The phenotype resulting from the knockout of a single allele needs investigation to align with observed human phenotypes and genetic data.

We thank the reviewer for recommendation. The phenotype of Capsl heterozygous mice at P5 showed no overt difference in vascular progression, vessel density and branchpoints with littermate wildtype controls (Fig.S4). The lack of pronounced phenotype in FEVR heterozygous mice may be due to different sensitivity between human and mice. A similar example is LRP5 mutations associated with FEVR. Heterozygous mutations in LRP5 were reported in FEVR patients in multiple populations. However, heterozygous Lrp5 mice exhibited no visible angiogenic phenotype (PMID: 18263894).

(4) The MYC pathway has been identified as influenced by CAPSL. Whether MYC downregulation is observed in the mouse model in vivo?

We thank the reviewer for recommendation. MYC expression was identified at both mRNA and protein level in Figure S8, and corresponding description was added in the manuscript at page 11.

**Reviewer #2 (Recommendations For The Authors):**
Minor comments:(1) While authors note that little is known about CAPSL protein function, more introductory detail about the protein (structure, domains intracellular localization etc) and additional discussion on potential mechanisms would aid the reader in interpreting the findings and model.

We thank the reviewer for recommendation. The subcellular localization of the CAPSL protein is distributed in both the nucleus and cytoplasm (https://www.proteinatlas.org/). The immunochemistry analysis confirmed that CAPSL protein is expressed in both the cell nucleus and cytoplasm (Fig.S1). And corresponding description was added in the manuscript at page 5.

(2) Pg 7 states that Capsl knockout mainly leads to "...defects in retinal vascular ECs rather than other vascular cells.". Consider rephrasing to describe "other vasculature-associated cells", as no vascular cells outside the retina were examined in the manuscript.

We thank the reviewer for recommendation. We rephrased the "...defects in retinal vascular ECs rather than other vascular cells." into "...defects in retinal vascular ECs rather than other vasculature-associated cells" at page 8.

(3) The manuscript is well written but contains numerous typos. E.g. "" (Pg 14), "MCY signaling axis" (figure 6 legend), "shCAPAL" (figure 5 K). Please correct these, and search carefully for others.

We are sorry for the careless mistakes we made, and we have checked the manuscript and correct these mistakes.

**Reviewer #3 (Recommendations For The Authors):**
The following are somewhat grammatical, but significant issues, that I feel should be addressed before making the pre-print final:(1) Perhaps the largest issue with the manuscript to me is whether CAPSL is an interesting candidate (as stated repeatedly) or causative of FEVR. Within the scope of what is feasible, this is a challenging problem. Since the publication of the pre-print, it would be great if another group independently reported the detection of mutations specifically in FEVR patients. That lacking, meaningful additions to the manuscript that I'd recommend are the inclusion of a paragraph on caveats of the study and reporting the allele frequencies based on public databases. As the authors know the data better than anyone and will have invested thought into the implications, they are the ones best positioned to alert the field to the study's limitations - amongst them- the factors that might practically distinguish whether CAPSL is a candidate or cause.

We thank the reviewer for recommendation. We will collect more samples from FEVR families and screen for other mutation sites within the CAPSL gene in further studies.

(2) It is unclear why the modeling with mice did not attempt to recapitulate the observations in humans, i.e., why were heterozygotes for a ubiquitous knockout not studied? Any data with heterozygotes, or ubiquitous alleles (which would be easier to generate than the strain studied) should be shared in the manuscript. If no such data exists, this reviewer would find it a worthwhile new experiment to add, but it is appreciated that new experiments are sometimes beyond the scope of what is possible. At the least, this would be worthwhile to discuss in the requested caveats paragraph of the discussion.

We thank the reviewer for recommendation. We evaluated the phenotype of Capsl heterozygous mice at P5, and the results showed no overt difference in vascular progression, vessel density and branchpoints with littermate wildtype controls (Fig.S4). The lack of pronounced phenotype in FEVR heterozygous mice may be due to different sensitivity between human and mice. For example, heterozygous Lrp5 mice exhibited no visible angiogenic phenotype (PMID: 18263894). Corresponding description was added in the manuscript at page 6.

(3) The statement in the Abstract "which provides invaluable information for genetic counseling and prenatal diagnosis of FEVR" should be toned down, better supported, or rephrased. This appears to be the 18th disease-associated gene for FEVR, with variants identified in 4 patients of the same ethnicity. In my opinion, the word "invaluable" is currently overstated.

We thank the reviewer for recommendation. We have changed "which provides invaluable information for genetic counseling and prenatal diagnosis of FEVR" into "which provides valuable information for genetic counseling and prenatal diagnosis of FEVR" in the abstract.

(4) The transcriptomic and proteomic data should be deposited into a public repository and accession numbers added to the manuscript.

We thank the reviewer for recommendation. We have uploaded the raw data of transcriptomic and proteomic to the Genome Sequence Archive (https://ngdc.cncb.ac.cn/gsa/browse/HRA010305) and the Genome Sequence Archive (https://www.ebi.ac.uk/pride/archive), respectively.

(5) The links to MYC are over-stated in the title "through the MYC axis", the abstract "CAPSL function causes FEVR through MYC axis", and the discussion "we demonstrated that the defects in CAPSL affect EC function by down-regulating the MYC signaling cascade". The links to MYC are entirely by association, there were no experiments testing that the transcriptomic and proteomic changes observed were determinative of the CAPSL-mediated phenotype. It seems appropriate to conjecture that these changes are important, but the above statements all need to be altered and conjectures need to be clearly identified as such.

We are sorry to overstate the link between CAPSL-mediated phenotype and MYC axis in the abstract and discussion sections, and we have altered the statements in these sections to make it more logical. For example, we changed “This study also reveals that compromised CAPSL function causes FEVR through MYC axis, shedding light on the potential involvement of MYC signaling in the pathogenesis of FEVR.” into “This study also reveals that compromised CAPSL function causes FEVR may through MYC axis, shedding light on the potential involvement of MYC signaling in the pathogenesis of FEVR.” in the abstract. And in the discussion we changed “…cause FEVR through inactivating MYC signaling, expanding FEVR-involved signaling pathway and providing a potential therapeutic target for the intervention of FEVR” to “…cause FEVR may through inactivating MYC signaling, expanding FEVR-involved signaling pathway and providing a potential therapeutic target for the intervention of FEVR”.

(6) Finally, I suggest that the following grammatical issues in the pre-print be corrected before making the pre-print final:

We have checked the manuscript and correct these mistakes.

(a) p2. Suggest rewriting the sentence "Nevertheless, the molecular mechanisms by which CAPSL regulates cell processes and signaling cascades have yet to be elucidated." The preceding sentences only state that CASPL is a candidate in another disease - the word "nevertheless" seems to reflect a logic that isn't described.

We have checked the manuscript and correct these mistakes.

(b) p5. Please correct the grammar "We, generated an inducible"

We corrected this mistake.

(c) p5. Suggest rephrasing "impairing CAPSL expression." The word "expression" is often used in reference to transcription. To avoid confusion, something such as "eliminating or reducing protein abundance" might be better.

We corrected this mistake.

(d) p6. Please correct the grammar "As expected, the radial vascular growth, as well as vessel density and vascular branching, are dramatically reduced in..." - note subject-verb agreement issue

We corrected this mistake.

(e) Figure 3 legend - correct "(A) Hyloaid vessels"

We corrected this mistake.